# CONTEXT-AWARE VIDEO INSTANCE SEGMENTATION

## ABSTRACT

In this paper, we introduce the Context-Aware Video Instance Segmentation (CAVIS), a novel framework designed to enhance instance association by integrating contextual information adjacent to each object. To efficiently extract and leverage this information, we propose the Context-Aware Instance Tracker (CAIT), which merges contextual data surrounding the instances with the core instance features to improve tracking accuracy. Additionally, we introduce the Prototypical Cross-frame Contrastive (PCC) loss, which ensures consistency in object-level features across frames, thereby significantly enhancing instance matching accuracy. CAVIS demonstrates superior performance over state-of-the-art methods on all benchmark datasets in video instance segmentation (VIS) and video panoptic segmentation (VPS). Notably, our method excels on the OVIS dataset, which is known for its particularly challenging videos. Source code: this anonymous URL

## 1 INTRODUCTION

Video Instance Segmentation (VIS) is a crucial task that involves segmenting and identifying individual objects within video sequences, applicable in a variety of fields including video understanding, autonomous driving, and video editing (Yang et al., 2019). VIS has seen considerable advancements, with developments in both online methods (Yang et al., 2019; Cao et al., 2020; Yang et al., 2021b; Huang et al., 2022; Wu et al., 2022c; Ying et al., 2023; Kim et al., 2024), which process videos frame-by-frame to adapt in real-time, and offline methods (Wang et al., 2021; Hwang et al., 2021; Wu et al., 2022b; Cheng et al., 2021a; Heo et al., 2022), which analyze entire videos to understand inter-frame dependencies thoroughly.

Recent advances have brought robust query-based segmentation architectures (Cheng et al., 2021b; 2022), designed to detect instance centers and cluster pixels into instance-specific groups within images. These modern VIS approaches increasingly employ instance center associations across frames to improve tracking accuracy. Techniques such as contrastive learning (Wu et al., 2022c; Ying et al., 2023) and transformer-based trackers (Heo et al., 2023; Zhang et al., 2023a) leverage the similarities between instance centers for consistent identification of objects across frames. However, challenges persist in scenarios with significant occlusions or when multiple similar objects are present, leading to potential tracking inaccuracies as shown in the top row of Fig. 1. While some strategies (Heo et al., 2022; 2023) attempt to use instance features for tracking across segments or entire videos, difficulties remain.

To address this issue, we propose Context-Aware Video Instance Segmentation (CAVIS), a novel framework designed to improve object identification by incorporating contextual information surrounding each instance into the tracking process. This approach draws from insights in neuroscience and cognitive science (Bar, 2004; Oliva & Torralba, 2007), emphasizing the importance of contextual cues in human perception for deciphering complex scenes and resolving visual ambiguities. An example of the practical application of this principle is shown in Fig. 1, where recognizing a person is riding a bicycle, rather than just identifying the bicycle, greatly enhances the accuracy of object identification.

To achieve this, we design the Context-Aware Instance Tracker (CAIT), featuring advanced modules for extracting and matching context-aware instance features. The context-aware instance feature extractor combines the contextual information at the object's boundary with the core features of each instance. Then, we incorporate these context-aware instance features into a transformer-based tracking architecture (Wu et al., 2022b; Heo et al., 2023; Zhang et al., 2023a), enhanced by our

Figure 1: **Importance of contextual information**. Comparative results showing the state-of-the-art model (Zhang et al., 2023b) (Top) and CAVIS (Bottom). The frame on the left precedes the right by four frames, during which an occlusion takes place. The standard model, lacking contextual data, fails to consistently track the same bicycle post-occlusion, while CAVIS effectively maintains accurate instance tracking.

novel context-aware cross-attention mechanism. This adjustment allows for the precise utilization of detailed contextual nuances within each scene.

Furthermore, we introduce the Prototypical Cross-frame Contrastive (PCC) loss to ensure temporal consistency across frames on high-level feature maps. Tracking methods in video tasks have predominantly focused on object feature representation learning (Wu et al., 2022c; Fischer et al., 2023; Ying et al., 2023; Li et al., 2023c), with recent studies exploring pixel-level representation learning on low-level feature maps (Kim et al., 2025). Building on this context, our PCC loss emphasizes high-level feature maps, inspired by the observation that object features and high-level feature maps are closely linked, as their similarity drives mask predictions. By constructing instance-wise prototypes from high-level feature maps, this loss maintains frame-to-frame consistency, enhancing training efficiency and ensuring robust performance in dynamic environments.

Our extensive testing shows that CAVIS significantly outperforms existing state-of-the-art methods across major video segmentation benchmarks, including YTVIS19 (Yang et al., 2019), YTVIS21 (Yang et al., 2021a), OVIS (Qi et al., 2022), and VIPSeg (Miao et al., 2021), particularly excelling on OVIS dataset that include complex video sequences. Our contributions to the field are manifold and can be summarized as follows:

1. We present Context-Aware Instance Tracker (CAIT), a novel framework designed to extract context-aware instance features and utilize them for enhanced instance matching.

2. We propose a Prototypical Cross-frame Contrative (PCC) loss that enhances the learning of instance matching by ensuring consistency in object-level features across frames.

3. Our model demonstrates robustness in challenging videos environments, establishing state-of-the-art performance in Video Instance Segmentation and Video Panoptic Segmentation.

## 2  RELATED WORKS

**Video Instance Segmentation.** VIS methods learn to associate features frame-to-frame based on instance segmentation architectures. The pioneering MaskTrack R-CNN (Yang et al., 2019) integrates a tracking head into Mask R-CNN (He et al., 2017), utilizing heuristic cues for instance association Following advancements include SipMask (Cao et al., 2020) and CrossVIS (Yang et al., 2021b), which enhance temporal links through cross-frame learning. IDOL(Wu et al., 2022c) a contrastive learning approach with query-based architectures (Zhu et al., 2020), boosting online method performance. Conversely, offline approaches like VisTR (Wang et al., 2021) and Seqformer (Wu et al., 2022b) use the entire video for mask trajectory predictions, with VisTR applying DETR (Carion et al., 2020) at the clip level and Seqformer aggregating temporal information via inter-frame queries. Innovations

like IFC (Hwang et al., 2021) and TeViT (Yang et al., 2022) improve efficiency by adjusting attention mechanisms within transformer architectures.

**Advancements in Query-based Networks.** Strong query-based segmentation networks have become prevalent in current VIS methods, with many relying on Mask2Former (Cheng et al., 2022) as their foundation. MinVIS (Huang et al., 2022) achieves tracking through simple post-processing based on cosine similarity between object features, without video learning. VITA (Heo et al., 2022) temporally associates frame-level queries to find instance prototypes within a video. GenVIS (Heo et al., 2023) adopts object association approach of VITA and designs a tracking network at the sub-clip level. Inspired by SimCLR (Chen et al., 2020), CTVIS (Ying et al., 2023) utilizes contrastive learning with a larger number of frames for comprehensive frame association. DVIS (Zhang et al., 2023a) introduces a decoupled framework for VIS, dividing it into segmentation, tracking, and refinement tasks, thereby enabling efficient and effective learning.

**Object Tracking with Additional Cues.** Tracking methodologies have been developed across various domains, including video object segmentation (VOS) (Xu et al., 2018; Oh et al., 2019; Cheng et al., 2021c), multiple object tracking (MOT) (Milan et al., 2016; Bergmann et al., 2019; Zhou et al., 2020; Zhang et al., 2021), and VIS. Despite advancements, many challenging cases persist, prompting research into object association with supplementary data. Early approaches leverage spatial-temporal information such as geometric relation between adjacent frames (Tang et al., 2017) and aggregated object features of previous frames (Xu et al., 2019). BeyondPixel (Sharma et al., 2018) improves inter-frame object matching by proposing a new cost that captures 3D pose and shape based on monocular geometry. BATMAN (Yu et al., 2022) combines optical flow and object query features to encode motion and appearance information into bilateral space. CAROQ (Choudhuri et al., 2023) employs a context feature defined as a memory bank of multi-level image features extracted by a pixel decoder. However, such a full-context-based approach can lead to increased complexity and memory limitations. In contrast, our method takes a memory-efficient approach by focusing on the surrounding features of each object during tracking, enabling effective object matching.

## 3 PRELIMINARY

This section offers a concise introduction to the fundamentals of a query-based instance segmentation pipeline and outlines a VIS approach that incorporates contrastive learning (Wu et al., 2022c; Li et al., 2023b; Ying et al., 2023; Li et al., 2023c).

### 3.1 QUERY-BASED INSTANCE SEGMENTATION

Modern VIS methods adopt a query-based instance segmentation pipeline (Cheng et al., 2021b; 2022), including three main components: a backbone encoder, a pixel decoder, and a transformer decoder. The backbone encoder and pixel decoder are responsible for extracting multi-scale feature maps from the input image. The transformer decoder employs object queries—sequences of latent vectors—as initial guesses for object centers and utilizes these features to generate object-level features. These queries undergo refinement through multiple transformer blocks via a cross-attention mechanism between the object queries and the feature maps. The refined instance features are then used for classification and segmentation tasks through respective prediction heads. Typically, the number of object queries, $N$, exceeds the actual number of objects, $N_{GT}$, present in the image. Traditionally, the process involves finding a permutation of $N$ elements, $\sigma \in \mathfrak{S}_N$, that optimally assigns the prediction set $\{\hat{y}_i\}_{i=1}^{N}$ to maximize total similarity to the ground truth (GT) set $\{y_i\}_{i=1}^{N_{GT}}$. This is achieved by minimizing a pair-wise matching cost $\mathcal{L}_{\text{Match}}$, as defined in (Cheng et al., 2021a):

$$\hat{\sigma} = \underset{\sigma \in \mathfrak{S}_N}{\arg\min} \sum_{k=1}^{N_{GT}} \mathcal{L}_{\text{Match}}\left(y_k, \hat{y}_{\sigma(k)}\right). \tag{1}$$

The network is trained with an objective function, $\mathcal{L}_{\text{Inst}}$, which consists of a categorical loss ($\mathcal{L}_{\text{Cls}}$), a binary cross-entropy loss for masks ($\mathcal{L}_{\text{Bce}}$), and a dice loss ($\mathcal{L}_{\text{Dice}}$) with the weights $\lambda_{\text{Cls}}$, $\lambda_{\text{Bce}}$, and $\lambda_{\text{Dice}}$ balancing the contributions of each loss component as follows:

$$\mathcal{L}_{\text{Inst}} = \lambda_{\text{Cls}}\mathcal{L}_{\text{Cls}} + \lambda_{\text{Bce}}\mathcal{L}_{\text{Bce}} + \lambda_{\text{Dice}}\mathcal{L}_{\text{Dice}}. \tag{2}$$

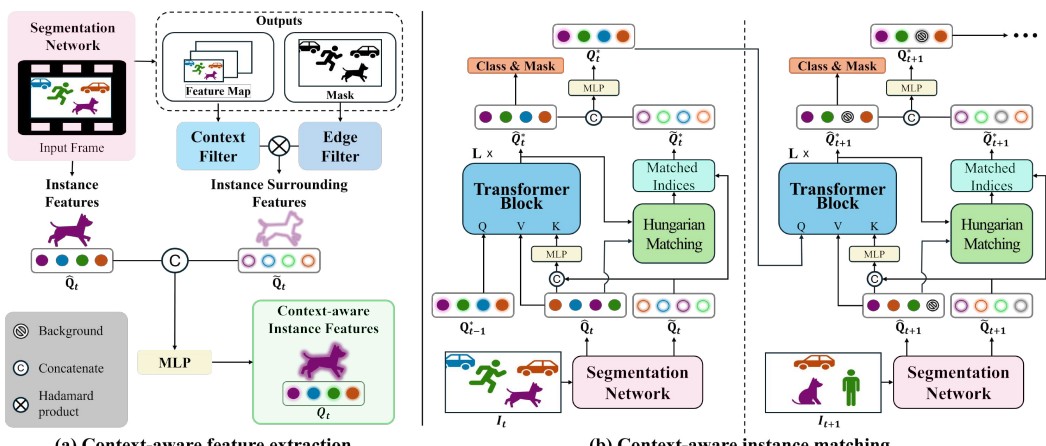

Figure 2: **Overview of CAVIS**. (a) The extraction of **context-aware instance features** from the output of an instance segmentation network. (b) CAVIS pipeline through **context-aware instance matching**. This includes the organization of the surrounding instance features $\tilde{Q}_t$, facilitated by Hungarian matching between the ordered instance features $\hat{Q}_t^*$ and unordered instance features $\hat{Q}_t$.

The loss is used to train VIS framework with the frame-wise matching relation $\hat{\sigma}^t$ as follows:

$$\mathcal{L}_{\text{VIS}} = \sum_{t=1}^{T} \sum_{n=1}^{N_{GT}} \mathcal{L}_{\text{Inst}} \left( y_n^t, \hat{y}_{\hat{\sigma}^t(n)}^t \right). \tag{3}$$

### 3.2 CONTRASTIVE LEARNING FOR VIS

In query-based architectures, the order of instance features effectively serves as the identity of each object. By aligning the sequence of instance features across frames, we can facilitate object tracking. Since instance features represent specific objects, inter-frame feature association is used for this alignment. To enhance the robustness of instance features for matching objects between frames, the following contrastive loss is integrated within the VIS framework (Wu et al., 2022c):

$$\mathcal{L}_{\text{Emb}}(v_t) = -\log \frac{\sum_{k^+ \in \mathbf{K}_{v_t}^+} \exp\left(v_t \cdot k^+\right)}{\sum_{k^+ \in \mathbf{K}_{v_t}^+} \exp\left(v_t \cdot k^+\right) + \sum_{k^- \in \mathbf{K}_{v_t}^-} \exp\left(v_t \cdot k^-\right)},$$

$$= \log\left[1 + \sum_{k^+ \in \mathbf{K}_{v_t}^+} \sum_{k^- \in \mathbf{K}_{v_t}^-} \exp\left(v_t \cdot k^- - v_t \cdot k^+\right)\right], \quad \forall t \in \{1, ..., T\}, \tag{4}$$

where $\mathbf{K}_{v_t}^+$ represents the sets of positive embeddings corresponding to the same object as $v_t$ from frames other than the $t$-th frame, while $\mathbf{K}_{v_t}^-$ includes negative embeddings featuring characteristics of objects different from that of $v_t$.

## 4 METHOD

This section describes our Context-Aware Video Instance Segmentation (CAVIS) whose overall pipeline is illustrated in Fig. 2. Our CAVIS consists of two key components: Context-Aware Instance Tracker (CAIT) and Prototypical Cross-frame Contrastive (PCC) loss, which are detailed in Sec. 4.1 and Sec. 4.2, respectively. We describe the training losses for each network in Sec. 4.3. We provide a notation table in Tab. 1 for better readability.

| Symbol | Description | Symbol | Description |
|--------|-------------|--------|-------------|
| $\hat{Q}$ | : instance features | $M$ | : mask predictions |
| $\tilde{Q}$ | : instance surrounding features | $\acute{M}$ | : boundary scores processed from $M$ |
| $Q$ | : context-aware instance features | $F$ | : the last feature maps from pixel decoder |
| $Q^*$ | : aligned context-aware instance features | $\bar{F}$ | : feature maps processed by average filter |

Table 1: Notations used in our method.

## 4.1 CONTEXT-AWARE INSTANCE TRACKER

### 4.1.1 CONTEXT-AWARE FEATURE EXTRACTION

Following the method outlined in previous VIS studies (Huang et al., 2022; Heo et al., 2022; 2023; Ying et al., 2023; Zhang et al., 2023a), we employ Mask2Former (Cheng et al., 2022) as our segmentation network $\mathcal{S}$. This framework ingests a series of input frames $\{I_t\}_{t=1}^T$, with $T$ denoting the total number of frames. It extracts feature maps $F$, identifies instance features $\hat{Q}$, and computes both classification scores $O$ and generates segmentation masks $M$ as follows:

$$\left\{F_t, \hat{Q}_t, O_t, M_t\right\}_{t=1}^T = \mathcal{S}\left(\{I_t\}_{t=1}^T\right),$$
$$F_t \in \mathbb{R}^{H \times W \times C}, \; \hat{Q}_t \in \mathbb{R}^{N \times C}, \; O_t \in \mathbb{R}^{N \times K}, \; M_t \in \mathbb{R}^{N \times H \times W}, \tag{5}$$

where $H, W,$ and $C$ denote the height, width, and channel dimensions of the feature maps, respectively. $N$ indicates the maximum number of detactable objects in a single frame, and $K$ signifies the number of object classes. We then extract the instance surrounding features $\tilde{Q}_t \in \mathbb{R}^{N \times C}$ capturing data around the object's boundaries essential for detailed context analysis as follows:

$$\tilde{Q}_t^n = \frac{\sum_{h=1}^H \sum_{w=1}^W \bar{F}_t^{\{h,w\}} * \mathbb{1}\left(\acute{M}_t^{\{n,h,w\}} > 0\right)}{\sum_{h=1}^H \sum_{w=1}^W \mathbb{1}\left(\acute{M}_t^{\{n,h,w\}} > 0\right)}, \quad \forall n = \{1, ..., N\}, \tag{6}$$
$$\text{where } \bar{F} = \text{Avg}(F), \; \acute{M} = \text{Lap}(M),$$

where $\text{Avg}(\cdot)$ denotes an average filtering process over spatial dimensions, $\text{Lap}(\cdot)$ signifies the application of a Laplacian filter, and $\mathbb{1}(\cdot)$ is the indicator function that tests for the presence of the object within the filtered mask. We employ the average filter specifically configured with a $9 \times 9$ kernel size.

Finally, we combine the core and surrounding features to further enhance instance representations. The context-aware instance feature $Q_t^n \in \mathbb{R}^C$ is generated by concatenating the core instance feature $\hat{Q}_t^n$ and the instance surrounding feature $\tilde{Q}_t^n$ along the channel dimension, and subsequently processing this combined feature through a multi-layer perceptron (MLP) as follows:

$$Q_t^n = \text{MLP}\left(\text{Concat}\left(\hat{Q}_t^n, \tilde{Q}_t^n\right)\right), \quad \forall n = \{1, ..., N\}. \tag{7}$$

The MLP is structured with three linear layers, each followed by a ReLU activation function. To promote the learning of discriminative context-aware instance features, we implement a contrastive loss specifically for these features. The context-aware contrastive loss, denoted as $\mathcal{L}_{\text{CTX}}$, leverages the established contrastive loss framework $\mathcal{L}_{\text{Emb}}$ detailed in Eq. (4) as follows:

$$\mathcal{L}_{\text{CTX}} = \sum_{t=1}^T \sum_{n=1}^{N_{GT}} \mathcal{L}_{\text{Emb}}\left(Q_t^{\hat{\sigma}^t(n)}\right), \tag{8}$$

where $\hat{\sigma}^t$ is a frame-wise matching relation as described in Eq. (1). This design enhances our ability to identify and track the same instances across the entire video. By not solely relying on instance centers, and instead utilizing context-aware instance embeddings, we can more accurately recognize instances throughout the video sequence.

### 4.1.2 CONTEXT-AWARE INSTANCE MATCHING

We introduce our context-aware tracking network $\mathcal{T}$, which employs a transformer-based tracking network (Heo et al., 2023; Zhang et al., 2023a) to learn associations across adjacent frames. The network comprises six transformer blocks, each featuring cross-attention, self-attention, and feed-forward layers. The conventional cross-attention mechanism, denoted as $\text{Attn}(Q, K, V)$, traditionally aligns the current unordered instance features, $\hat{Q}_t$, (serving as both the key and value), with the ordered instance features from the previous frame, $\hat{Q}_{t-1}^*$, (used as the query). To enhance accuracy, our model employs context-aware instance features, $Q_{t-1}^*$ and $Q_t$, as the query and key, respectively while we still use the instance features $\hat{Q}_t$ as value. This modification leads to a context-aware cross-attention mechanism, formulated as:

$$\text{Attn}(Q_{t-1}^*, Q_t, \hat{Q}_t) = \text{softmax}\left(\frac{Q_{t-1}^* \cdot (Q_t)^T}{\sqrt{C}}\right)\hat{Q}_t, \tag{9}$$

where $C$ is the channel dimensions of the feature maps. The aligned context-aware instance features $Q_t^*$ is built by concatenating the aligned instance features $\hat{Q}_t^*$ and the aligned instance surrounding features $\tilde{Q}_t^*$ same as in Eq. (7). We obtain the aligned instance surrounding features $\tilde{Q}_t^*$ by using Hungarian matching algorithm (Kuhn, 1955) on cosine similarity between $\hat{Q}_t$ and $\hat{Q}_t^*$ as follows:

$$\tilde{Q}_t^{*\,\sigma_H(n)} = \tilde{Q}_t^n, \ \forall n = \{1, ..., N\}, \ \text{where } \sigma_H = \text{Hungarian}(\hat{Q}_t^*, \hat{Q}_t), \ \sigma_H \in \mathbb{R}^N. \tag{10}$$

This process is similarly applied to the aligned context-aware features $Q_{t-1}^*$ for the previous frame.

### 4.2 PROTOTYPICAL CROSS-FRAME CONTRASTIVE LOSS

Current VIS methodologies emphasize the importance of object feature representation learning, especially in tracking tasks where matching object features across frames is crucial. In a query-based segmenter, object features $\hat{Q}_t^n \in \mathbb{R}^C$ are semantically correlated with each pixel embedding of the feature map $F_t^{\{h,w\}} \in \mathbb{R}^C$ from the pixel decoder, as they are used for mask prediction. This ensures consistent feature representation within object-containing regions and introduces an intra-frame constraint that reflects similarity within the feature map. By examining the inter-frame relationships of pixel embeddings, our method improves object association, essential for contrastive learning methods that strive to differentiate between similar and dissimilar objects across frames.

Given the memory-intensive nature of maintaining individual pixel consistency, we introduce Prototypical Cross-frame Contrastive (PCC) loss. This loss $\mathcal{L}_{\text{PCC}}$ maintains frame-to-frame consistency of pixel embeddings for each instance feature by constructing instance-wise prototypes from predicted masks, defined as follows:

$$\mathcal{L}_{\text{PCC}} = \sum_{t=1}^{T}\sum_{n=1}^{N_{GT}} \mathcal{L}_{\text{Emb}}\left(\eta_t^{\hat{\sigma}^t(n)}\right), \ \ \eta_t^n = \frac{\sum_{h=1}^{H}\sum_{w=1}^{W} F_t^{\{h,w\}} * \mathbb{1}\left(M_t^{\{h,w\}} == 1\right)}{\sum_{h=1}^{H}\sum_{w=1}^{W} \mathbb{1}\left(M_t^{\{h,w\}} == 1\right)}. \tag{11}$$

### 4.3 TRAINING LOSS

To train the segmentation network $\mathcal{S}$, we implement an objective function that incorporates the standard video instance segmentation loss $\mathcal{L}_{\text{VIS}}$ in Eq. (3), context-aware constrative loss $\mathcal{L}_{\text{CTX}}$ in Eq. (8) and Prototypical Cross-frame Contrastive (PCC) loss in Eq. (11) as follows:

$$\mathcal{L}_{\mathcal{S}} = \mathcal{L}_{\text{VIS}} + \lambda_{\text{CTX}}\mathcal{L}_{\text{CTX}} + \lambda_{\text{PCC}}\mathcal{L}_{\text{PCC}}, \tag{12}$$

where $\lambda_{\text{CTX}}$ and $\lambda_{\text{PCC}}$ are the weights assigned to balance these losses.

To train the tracking network $\mathcal{T}$, we calculate the matching cost only for objects that appear for the first time to ensure consistent video-level matching pairs, as implemented in prior work (Zhang et al., 2023a). The objective function $\mathcal{L}_{\mathcal{T}}$ incorporating the matching relation $\hat{\sigma}_{\text{con}}$ is defined as follows:

$$\mathcal{L}_{\mathcal{T}} = \sum_{t=1}^{T}\sum_{n=1}^{N_{GT}} \mathcal{L}_{\text{Inst}}\left(y_n^t, \hat{y}_{\hat{\sigma}_{\text{con}}(n)}^t\right), \ \ \hat{\sigma}_{\text{con}} = \underset{\sigma \in \mathfrak{S}_N}{\arg\min}\sum_{k=1}^{N_{GT}} \mathcal{L}_{\text{Match}}\left(y_k^{f(k)}, \hat{y}_{\sigma(k)}^{f(k)}\right), \tag{13}$$

where $f(k)$ denotes the frame in which the $k$-th instance first appears.

Table 2: Comparisons on the validation sets of YouTube-VIS 2019, 2021, and OVIS datasets. The best and second-best scores are highlighted in **red** and **blue**, respectively. † denotes the model trained with the temporal refiner (Zhang et al., 2023a). Rows in cyan indicate comparisons with a top-performing model.

| Methods | OVIS | | | | | YTVIS19 | | | | | YTVIS21 | | | | |
|---|---|---|---|---|---|---|---|---|---|---|---|---|---|---|---|
| | AP | AP$_{50}$ | AP$_{75}$ | AR$_1$ | AR$_{10}$ | AP | AP$_{50}$ | AP$_{75}$ | AR$_1$ | AR$_{10}$ | AP | AP$_{50}$ | AP$_{75}$ | AR$_1$ | AR$_{10}$ |
| **ResNet-50** | | | | | | | | | | | | | | | |
| MaskTrack R-CNN (Yang et al., 2019) | 10.8 | 25.3 | 8.5 | 7.9 | 14.9 | 30.3 | 51.1 | 32.6 | 31.0 | 35.5 | 28.6 | 48.9 | 29.6 | 26.5 | 33.8 |
| SipMask (Cao et al., 2020) | 10.2 | 24.7 | 7.8 | 7.9 | 15.8 | 33.7 | 54.1 | 35.8 | 35.4 | 40.1 | 31.7 | 52.5 | 34.0 | 30.8 | 37.8 |
| IFC (Hwang et al., 2021) | 13.1 | 27.8 | 11.6 | 9.4 | 23.9 | 41.2 | 65.1 | 44.6 | 42.3 | 49.6 | 35.2 | 55.9 | 37.7 | 32.6 | 42.9 |
| CrossVIS (Yang et al., 2021b) | 14.9 | 32.7 | 12.1 | 10.3 | 19.8 | 36.3 | 56.8 | 38.9 | 35.6 | 40.7 | 34.2 | 54.4 | 37.9 | 30.4 | 38.2 |
| EfficientVIS (Wu et al., 2022a) | - | - | - | - | - | 37.9 | 59.7 | 43.0 | 40.3 | 46.6 | 34.0 | 57.5 | 37.3 | 33.8 | 42.5 |
| SeqFormer (Wu et al., 2022b) | 15.1 | 31.9 | 13.8 | 10.4 | 27.1 | 47.4 | 69.8 | 51.8 | 45.5 | 54.8 | 40.5 | 62.4 | 43.7 | 36.1 | 48.1 |
| VISOLO (Han et al., 2022) | 15.3 | 31.0 | 13.8 | 11.1 | 21.7 | 38.6 | 56.3 | 43.7 | 35.7 | 42.5 | 36.9 | 54.7 | 40.2 | 30.6 | 40.9 |
| Mask2Former-VIS (Cheng et al., 2021a) | 17.3 | 37.3 | 15.1 | 10.5 | 23.5 | 46.4 | 68.0 | 50.0 | - | - | 40.6 | 60.9 | 41.8 | - | - |
| VITA (Heo et al., 2022) | 19.6 | 41.2 | 17.4 | 11.7 | 26.0 | 49.8 | 72.6 | 54.5 | 49.4 | 61.0 | 45.7 | 67.4 | 49.5 | 40.9 | 53.6 |
| MinVIS (Huang et al., 2022) | 25.0 | 45.5 | 24.0 | 13.9 | 29.7 | 47.4 | 69.0 | 52.1 | 45.7 | 55.7 | 44.2 | 66.0 | 48.1 | 39.2 | 51.7 |
| CAROQ (Choudhuri et al., 2023) | 25.8 | 47.9 | 25.4 | 14.2 | 33.9 | 46.7 | 70.4 | 50.9 | 45.7 | 55.9 | 43.3 | 64.9 | 47.1 | 39.3 | 52.7 |
| IDOL (Wu et al., 2022c) | 28.2 | 51.0 | 28.0 | 14.5 | 38.6 | 49.5 | 74.0 | 52.9 | 47.7 | 58.7 | 43.9 | 68.0 | 49.6 | 38.0 | 50.9 |
| DVIS (Zhang et al., 2023a) | 30.2 | 55.0 | 30.5 | 14.5 | 37.3 | 51.2 | 73.8 | 57.1 | 47.2 | 59.3 | 46.4 | 68.4 | 49.6 | 39.7 | 53.5 |
| TCOVIS (Li et al., 2023a) | 35.3 | 60.7 | **36.6** | 15.7 | 39.5 | 52.3 | 73.5 | 57.6 | 49.8 | 60.2 | 49.5 | 71.2 | 53.8 | 41.3 | 55.9 |
| CTVIS (Ying et al., 2023) | 35.5 | **60.8** | 34.9 | 16.1 | **41.9** | 55.1 | **78.2** | 59.1 | **51.9** | **63.2** | 50.1 | 73.7 | 54.7 | 41.8 | **59.5** |
| GenVIS (Heo et al., 2023) | 35.8 | **60.8** | 36.2 | **16.3** | 39.6 | 50.0 | 71.5 | 54.6 | 49.5 | 59.7 | 47.1 | 67.5 | 51.5 | 41.6 | 54.7 |
| VISAGE (Kim et al., 2025) | **36.2** | 60.3 | 35.3 | 16.1 | 40.3 | **55.1** | 78.1 | **60.6** | 51.0 | 62.3 | **51.6** | **73.8** | **56.1** | **43.6** | 59.3 |
| Ours | 37.6 | 63.4 | 38.2 | 16.5 | 43.5 | 55.7 | 78.3 | 61.7 | 51.5 | 63.3 | 50.5 | 74.1 | 54.9 | 42.6 | 58.5 |
| **Swin-L** | | | | | | | | | | | | | | | |
| SeqFormer (Wu et al., 2022b) | - | - | - | - | - | 59.3 | 82.1 | 66.4 | 51.7 | 64.4 | 51.8 | 74.6 | 58.2 | 42.8 | 58.1 |
| Mask2Former-VIS (Cheng et al., 2021a) | 25.8 | 46.5 | 24.4 | 13.7 | 32.2 | 60.4 | 84.4 | 67.0 | - | - | 52.6 | 76.4 | 57.2 | - | - |
| VITA (Heo et al., 2022) | 27.7 | 51.9 | 24.9 | 14.9 | 33.0 | 63.0 | 86.9 | 67.9 | 56.3 | 68.1 | 57.5 | 80.6 | 61.0 | 47.7 | 62.6 |
| CAROQ (Choudhuri et al., 2023) | 38.2 | 60.7 | 39.5 | 17.7 | 44.1 | 61.4 | 82.8 | 68.6 | 55.2 | 68.1 | 54.5 | 75.4 | 60.5 | 45.5 | 61.4 |
| MinVIS (Huang et al., 2022) | 39.4 | 61.5 | 41.3 | 18.1 | 43.3 | 61.6 | 83.3 | 68.6 | 54.8 | 66.6 | 55.3 | 76.6 | 62.0 | 45.9 | 60.8 |
| IDOL (Wu et al., 2022c) | 40.0 | 63.1 | 40.5 | 17.6 | 46.4 | 64.3 | 87.5 | 71.0 | 55.6 | 69.1 | 56.1 | 80.8 | 63.5 | 45.0 | 60.1 |
| GenVIS (Heo et al., 2023) | 45.2 | 69.1 | 48.4 | **19.1** | 46.2 | 64.0 | 84.9 | 68.3 | 56.1 | 69.4 | 59.6 | 80.9 | 65.8 | **48.7** | 65.0 |
| DVIS (Zhang et al., 2023a) | 45.9 | 71.1 | 48.3 | 18.5 | 51.5 | 63.9 | 87.2 | 70.4 | 56.2 | 69.0 | 58.7 | 80.4 | 66.6 | 47.5 | 64.6 |
| TCOVIS (Li et al., 2023a) | 46.7 | 70.9 | **49.5** | **19.1** | 50.8 | 64.1 | 86.6 | 69.5 | 55.8 | 69.0 | **61.3** | 82.9 | 68.0 | **48.6** | 65.1 |
| CTVIS (Ying et al., 2023) | **46.9** | **71.5** | 47.5 | **19.1** | **52.1** | **65.6** | **87.7** | **72.2** | **56.5** | **70.4** | 61.2 | **84.0** | **68.8** | 48.0 | **65.8** |
| Ours | 48.6 | 74.0 | 52.5 | 19.5 | 53.3 | 66.0 | 89.5 | 73.3 | 56.8 | 71.4 | 61.1 | 84.1 | 69.2 | 48.2 | 66.3 |
| **ViT-L** | | | | | | | | | | | | | | | |
| MinVIS (Huang et al., 2022) | 42.9 | 65.7 | 45.4 | 19.8 | 46.5 | 65.6 | 85.4 | 72.7 | 57.5 | 70.6 | 59.2 | 79.9 | 66.7 | 47.8 | 64.1 |
| DVIS++ (Zhang et al., 2023b) | **49.6** | **72.5** | **55.0** | **20.8** | **54.6** | **67.7** | **88.8** | **75.3** | **57.9** | **73.7** | **62.3** | **82.7** | **70.2** | **49.5** | **68.0** |
| Ours | 53.2 | 75.9 | 59.1 | 20.9 | 58.2 | 68.9 | 89.3 | 76.2 | 58.3 | 73.6 | 64.6 | 85.6 | 72.5 | 49.5 | 69.3 |
| DVIS++† (Zhang et al., 2023b) | **53.4** | **78.9** | **58.5** | **21.1** | **58.7** | **68.3** | **90.3** | **76.1** | **57.8** | **73.4** | **63.9** | **86.7** | **71.5** | **48.8** | **69.5** |
| Ours† | 57.1 | 82.6 | 63.5 | 21.2 | 61.8 | 69.4 | 90.9 | 77.2 | 58.3 | 74.7 | 65.3 | 87.3 | 73.2 | 49.7 | 70.3 |

## 5 EXPERIMENTS

We evaluate CAVIS on two major tasks: video instance segmentation (VIS) and video panoptic segmentation (VPS) on four benchmark datasets recognized for their challenges and prevalence in the research community: YouTubeVIS-2019 (Yang et al., 2019), YouTubeVIS-21 (Yang et al., 2021a), OVIS (Qi et al., 2022), and VIPSeg (Miao et al., 2021). For VIS, performance metrics include average precision (AP) and average recall (AR) as established in previous studies (Yang et al., 2019). In the realm of VPS (Kim et al., 2020), we further examine our model's capabilities using the Segmentation and Tracking Quality (STQ) metric, and the Video Panoptic Quality (VPQ) metric.

### 5.1 IMPLEMENTATION DETAILS

We employ Mask2Former (Cheng et al., 2022) as our segmentation network, utilizing three distinct backbone encoders: ResNet-50 (He et al., 2016), Swin-L (Liu et al., 2021), and ViT-L (Dosovitskiy et al., 2021). The ResNet-50 and Swin-L backbones are initialized with parameters pre-trained on the COCO dataset (Lin et al., 2014), while the ViT-L backbone uses initialization parameters from DINOv2 (Oquab et al., 2023). Additionally, for the ViT-L backbone, we employ a memory-efficient version of VIT-Adapter (Chen et al., 2022), aligning with recent advancements in network efficiency (Zhang et al., 2023b). Further details are described in Sec. A.2.2.

### 5.2 COMPARISON TO STATE-OF-THE-ART METHODS

**Video Instance Segmentation (VIS).** We benchmark CAVIS against leading methods on three established VIS datasets, as detailed in Tab. 2. CAVIS sets a new state-of-the-art, outperforming the previous top model, DVIS++, by margins of 1.1, 1.4, and 3.7 average precision (AP) points on YouTube-VIS2019, YouTube-VIS2021, and OVIS, respectively. Particularly noteworthy is CAVIS's performance on the OVIS dataset, where it significantly outstrips all competitors. This dataset is renowned for its diversity and the complexity of its video sequences. Fig. 3 illustrates how our model proficiently tracks objects even in scenarios marked by severe occlusion. This capability underscores

Table 3: Comparison on VIPSeg validation sets. 'Th' and 'St' denote 'things' and 'stuff' classes. †
denotes the model trained with the temporal refiner (Zhang et al., 2023a).

| Method | ResNet-50 | | | | ViT-L | | | |
|---|---|---|---|---|---|---|---|---|
| | VPQ | VPQ$^{Th}$ | VPQ$^{St}$ | STQ | VPQ | VPQ$^{Th}$ | VPQ$^{St}$ | STQ |
| VPSNet-SiamTrack(Woo et al., 2021) | 17.2 | 17.3 | 17.3 | 21.1 | - | - | - | - |
| VIP-Deeplab(Qiao et al., 2021) | 16.0 | 12.3 | 18.2 | 22.0 | - | - | - | - |
| Clip-PanoFCN(Miao et al., 2022) | 22.9 | 25.0 | 20.8 | 31.5 | - | - | - | - |
| Video K-Net (Li et al., 2022) | 26.1 | - | - | 31.5 | - | - | - | - |
| TarVIS(Athar et al., 2023) | 33.5 | 39.2 | 28.5 | 43.1 | - | - | - | - |
| Tube-Link(Li et al., 2023c) | 39.2 | - | - | 39.5 | - | - | - | - |
| Video-kMax (Shin et al., 2024) | 38.2 | - | - | 39.9 | - | - | - | - |
| DVIS (Zhang et al., 2023a) | 39.4 | 38.6 | 40.1 | 36.3 | - | - | - | - |
| DVIS++ (Zhang et al., 2023b) | 41.9 | 41.0 | 42.7 | 38.5 | 56.0 | 58.0 | 54.3 | 49.8 |
| Ours | 42.4 | 43.1 | 41.8 | 39.7 | 56.9 | 60.1 | 54.2 | 51.0 |
| DVIS † (Zhang et al., 2023a) | 43.2 | 43.6 | 42.8 | 42.8 | - | - | - | - |
| DVIS++ † (Zhang et al., 2023b) | **44.2** | **44.5** | **43.9** | **43.6** | **58.0** | **61.2** | **55.2** | **56.0** |
| Ours † | **45.3** | **47.5** | **43.4** | **45.3** | **58.5** | **63.1** | **54.5** | **56.1** |

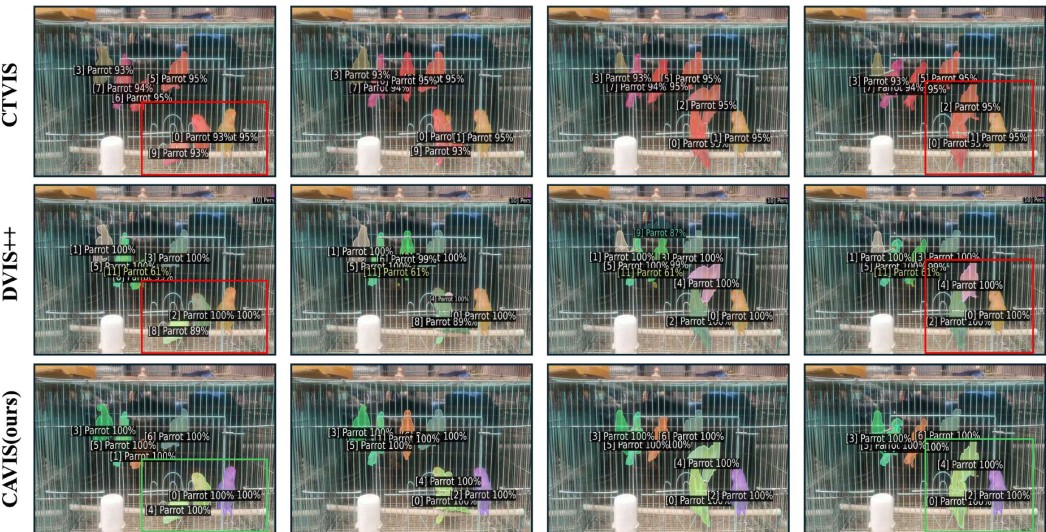

Figure 3: Qualitative comparisons of CAVIS (ours) against state-of-the-art methods: CTVIS (Ying
et al., 2023) and DVIS++ (Zhang et al., 2023b) on the OVIS dataset.

the strength of our context-aware video learning approach, which effectively leverages information
from surrounding objects for accurate instance matching, even under severe occlusion.

**Video Panoptic Segmentation (VPS).** In the realm of VPS, CAVIS also achieves the best perfor-
mance on the VIPSeg dataset as shown in Tab. 3. For the ResNet-50 backbone, it achieves 45.3
in both Video Panoptic Quality (VPQ) and Segmentation and Tracking Quality (STQ). For the
ViT-L backbone, the figures reach 58.5 VPQ and 56.1 STQ, demonstrating substantial advancements.
Specifically, our model shows significant gains in VPQ$^{Th}$—which assesses performance on 'thing'
classes—with increases of 3.0 and 1.9 for ResNet-50 and ViT-L backbones, respectively, over the
previous best models. These improvements highlight the versatility of our context-aware object
matching strategy across various video segmentation tasks.

## 5.3 ABLATION STUDY

We conduct ablation studies on the OVIS dataset (Qi et al., 2022) with the ResNet-50 (He et al., 2016)
backbone, detailed in Tab. 4. We use the same experimental setting as those in the main experiments.
To evaluate the segmentation network across these setups in Tab. 4-(a-c), we employ the minimal
post-processing method proposed by MinVIS (Huang et al., 2022).

Table 4: Ablation studies on each component of CAVIS.

(a) CAIT, PCC loss

| | Segmenter ($\mathcal{S}$) | | | Tracker ($\mathcal{T}$) | |
|---|---|---|---|---|---|
| | $\mathcal{L}_{\text{CTX}}$ | $\mathcal{L}_{\text{PCC}}$ | AP | Context-aware matching | AP |
| (i) | | | 26.4 | | 33.2 |
| (ii) | | ✓ | 28.1 | | 34.2 |
| (iii) | ✓ | | 29.7 | | 34.8 |
| (iv) | ✓ | | 29.7 | ✓ | 37.2 |
| (v) | ✓ | ✓ | **30.0** | | 35.3 |
| (vi) | ✓ | ✓ | **30.0** | ✓ | **37.6** |

(b) Context filter size, the number of frames

| Metric: AP | | # of frames | | |
|---|---|---|---|---|
| | | 2 | 3 | 4 |
| Filter size | 3 × 3 | 27.3 | 27.3 | 27.2 |
| | 5 × 5 | 28.3 | 28.7 | 28.1 |
| | 7 × 7 | 28.7 | 29.3 | 28.1 |
| | 9 × 9 | 29.5 | **30.0** | 28.7 |
| | 11 × 11 | 28.7 | 29.1 | 28.0 |

(c) Context filter type in $\mathcal{S}$

| Metric | Context filter type in $\mathcal{S}$ | | | |
|---|---|---|---|---|
| | Average | Max | Median | Learnable |
| AP | **30.0** | 29.4 | 29.6 | 28.7 |

(d) Context alignment in $\mathcal{T}$

| Context alignment in $\mathcal{T}$ | |
|---|---|
| ✗ | ✓ |
| 33.2 | **37.6** |

(e) Value for CAIT

| Value for CAIT | |
|---|---|
| $Q$ | $\hat{Q}$ |
| 36.8 | **37.6** |

**Ablation study on technical contributions of CAVIS.** We conduct a series of experiments in Tab. 4-(a) to demonstrate the effectiveness of our key components: the context-aware instance tracker (CAIT) and prototypical cross-frame contrastive (PCC) loss.

Tab. 4-(a) includes six experiments (i- vi), where experiment (i) present our baseline performance of retraining MinVIS (Huang et al., 2022) and DVIS (Zhang et al., 2023a) to match our settings. For segmentation netowrk ($\mathcal{S}$), experiments (iii- iv) show that implementing contrastive learning with context-aware instance features results in a notable +3.3 AP improvement over the baseline MinVIS. Further performance boosts are noted when PCC is introduced in experiments (v- vi), which records the highest performance of 30.0 AP by utilizing both $\mathcal{L}_{\text{CTX}}$ and $\mathcal{L}_{\text{PCC}}$. This highlights the synergistic effect of integrating context-aware tracking with cross-frame contrastive loss, significantly enhancing the system's accuracy and effectiveness.

We also evaluate the tracking network ($\mathcal{T}$) by initially using each fixed pre-trained segmentation network listed in Tab. 4-(i- vi). Comparing setups (v) and (vi), our newly designed context-aware cross-attention improves performance by +2.3 AP over the standard cross-attention, achieving 37.6 AP. These findings validate the efficacy of the context-aware feature, confirming its significant advantages for instance matching in complex video scenarios.

**Context filter.** Given that images feature objects at various scales, identifying the optimal receptive field size that functions effectively across different scenarios is essential. Our experiments, detailed in Tab. 4-(b), explore the effects of varying context filter sizes from 3 to 11. The optimal performance is achieved with a filter size of 9; larger sizes led to decreased performance, suggesting that excessively large receptive fields may detract from effective object matching by homogenizing the context features across all objects. Extended analysis on this aspect is provided in Sec. A.3. Further investigation into different types of context filters shown in Tab. 4-(c) reveals that the average filter, which evenly reflects surrounding information, offers a well-defined benefit. In contrast, the learnable filter, which lacks specific directives on characterizing surrounding information, performs similarly to scenarios without enhanced context features, as demonstrated in Tab. 4-(a)-(ii). This indicates the importance of a clearly defined context filter in improving the segmentation and tracking accuracy.

**The number of adjacent frames used during training.** Our fundamental assumption is that the surrounding information between adjacent frames remains relatively stable, thereby aiding object matching. Tab. 4-(b) details the performance comparison based on the number of adjacent frames used during training. Utilizing three frames results in the highest performance, achieving 30.0 AP. Increasing the number of frames decreases performance, likely due to larger gaps between sampled frames which lead to more significant changes in the surrounding information, thus complicating object matching. Consequently, we have found that using three frames optimizes training effectiveness.

**Context feature alignment.** The tracking network aligns instance features and produces outputs in varying orders, necessitating the precise alignment of context features for accurate matching

in subsequent frames. Misalignment of these features can lead to incorrect matching of context information for each object, significantly impacting performance. As demonstrated in Tab. 4-(d), misalignment results in a notable performance drop of 4.4 AP. This underscores the critical need for accurate alignment of context information to ensure robust object tracking performance.

**Value for context-aware instance matching.** Context-aware features ($Q$), which include information on instance features, could be used as the value in context-aware matching. However, during segmenter training, the instance features ($\hat{Q}$) specifically drive the segmentation prediction. Therefore, it is more effective to use instance features as the value for matching, leading to better performance, as shown in Tab. 4-(e).

## 6 COMPUTATIONAL COST

We compare the inference speed of our approach against recent state-of-the-art methods, GenVIS (Heo et al., 2023) and DVIS (Zhang et al., 2023a), to evaluate the computational cost. As shown in Tab. 5, the inference speeds were measured under identical conditions on a 2080ti GPU. Our method requires an additional time cost of 5.5ms and 6.7ms compared to GenVIS

| Method | Time (ms) | YTVIS19 (AP) |
|--------|-----------|--------------|
| GenVIS | 80.1 | 50.0 |
| DVIS | 78.9 | 51.2 |
| Ours | 85.6 | 55.7 |

Table 5: Inference speed.

and DVIS, respectively. However, this cost is justified by the performance gains of +5.7AP and +4.5AP, demonstrating a reasonable trade-off between increased computation and improved accuracy.

## 7 CONCLUSION

In this paper, we introduce Context-Aware Video Instance Segmentation (CAVIS), a pioneering framework designed to enhance the accuracy and reliability of object tracking in complex video scenarios by integrating contextual information surrounding each instance. The introduction of the Context-Aware Instance Tracker (CAIT) and the innovative Prototypical Cross-frame Contrastive (PCC) loss are central to CAVIS's effectiveness. CAIT leverages the surrounding context to enrich the core features of each instance, providing a more holistic view that significantly improves instance detection and segmentation under challenging conditions. Simultaneously, PCC loss ensures consistency of these enriched features across frames, reinforcing the temporal linkage between instances and enhancing the overall tracking robustness. Our experiments across multiple challenging benchmarks demonstrate that CAVIS significantly outperforms existing state-of-the-art methods, particularly excelling in scenarios that demand robust tracking capabilities. The integration of CAIT and PCC not only addresses the primary challenges of occlusions and motion but also effectively manages the presence of visually similar objects that can often mislead traditional VIS approaches.

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

# A APPENDIX

## A.1 LIMITATION

Video Instance Segmentation (VIS) is an advanced technology designed to perform segmentation and tracking concurrently, capturing the trajectories of individual instances within a video. While this technology has significant benefits, it also poses potential risks if misused, particularly in surveillance applications. Such misuse could lead to severe privacy infringements. It is important to note, however, that the dataset used in this study is a standard one within the VIS community and does not include any sensitive or personal information. This precaution helps mitigate the risk of our trained model being used for harmful purposes. Nonetheless, the potential for negative impacts should not be underestimated, and ethical considerations must guide the deployment of VIS technologies.

**Potential error in prediction.** Our model is designed to improve tracking accuracy by achieving precise object matching across frames rather than focusing on segmentation performance. Consequently, if the pretrained segmentation network produces inaccurate segmentation results, performance may decrease. However, even in scenarios with imprecise mask predictions, our proposed context-aware modeling can robustly track objects, as demonstrated in Fig. 4.

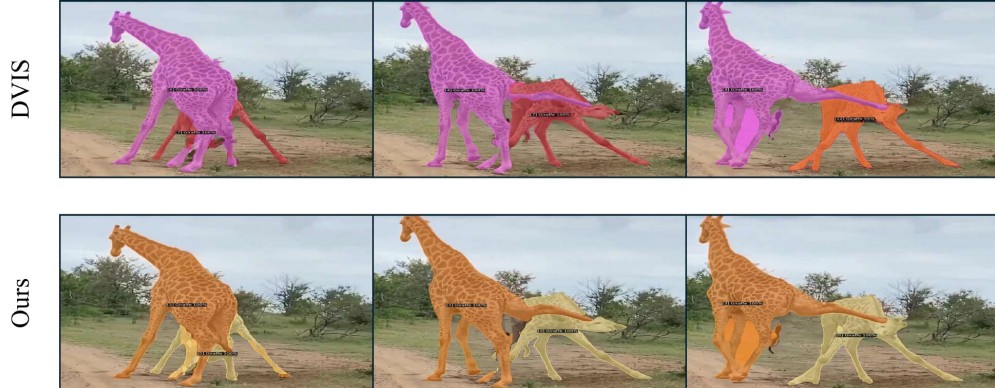

Figure 4: Potential error due to inaccurate mask predictions from the segmentation network.

## A.2 EXPERIMENTAL DETAILS

### A.2.1 DATASETS

**Youtube-VIS 2019 and 2021** YouTube-VIS was introduced by Yang et al. in their pioneering study on the VIS task (Yang et al., 2019). This dataset comprises high-resolution YouTube videos, categorized into 40 distinct classes. The 2019 version of the dataset includes 2,238 videos for training, 302 for validation, and 343 for testing (Yang et al., 2019). The 2021 update expands these numbers to 2,985, 421, and 453 videos for training, validation, and testing, respectively (Yang et al., 2021a). YouTube-VIS is utilized across various pixel-level video understanding tasks, including VIS, video semantic segmentation, and video object detection.

**OVIS** The OVIS dataset (Qi et al., 2022) presents a significant challenge with its frequent occlusions and a realistic representation of common everyday objects. This makes it highly relevant for real-world applications. OVIS videos are longer and contain more objects compared to those in YouTube-VIS, which increases the complexity of segmentation and tracking tasks. The dataset is organized into training, validation, and test sets, with 607, 140, and 154 videos, respectively.

**VIPSeg** VIPSeg (Miao et al., 2022) is a comprehensive Video Panoptic Segmentation dataset that includes 3,536 videos and 84,750 frames, annotated with pixel-level panoptic labels. Unlike earlier VPS datasets that primarily focus on street views, VIPSeg offers a broader range of challenges and practical scenarios. It features 232 diverse settings and is annotated with 58 'thing' classes and 66 'stuff' classes, making it one of the most diverse and challenging datasets available in the field.

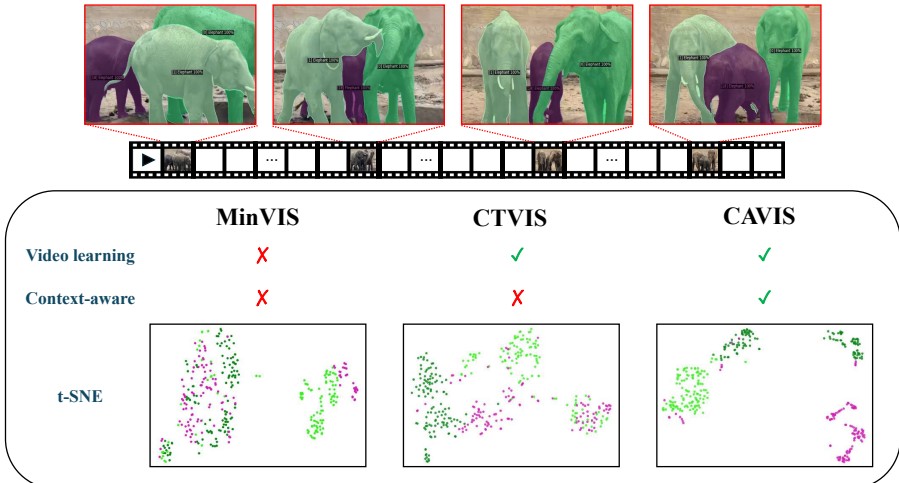

Figure 5: Visualization of object embeddings. Each point on the t-SNE (Van der Maaten & Hinton, 2008) plot represents the learned object embeddings. The three different colors of points indicate the embeddings of three different elephants throughout the entire video.

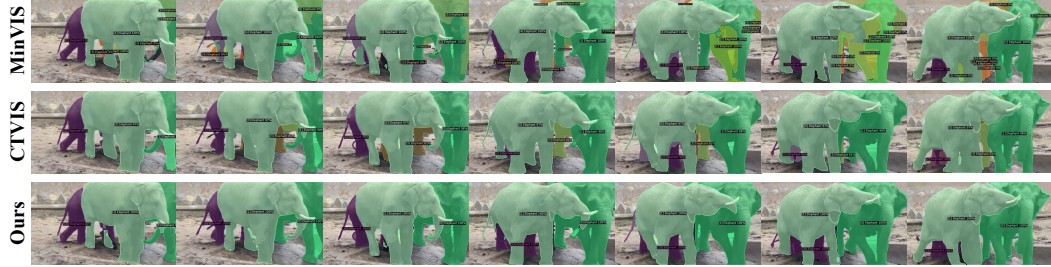

Figure 6: Comparison of VIS results for the video in Fig. 5. These results show that our model robustly tracks objects even in scenes with severe occlusions.

### A.2.2 IMPLEMENTATION

Our segmentation approach employs the Mask2Former architecture (Cheng et al., 2022), utilizing the officially recommended hyperparameters. For all experimental settings, we follow established practices by incorporating COCO joint training, as adopted in previous methodologies (Wu et al., 2022b; Heo et al., 2022; 2023; Ying et al., 2023; Zhang et al., 2023a). The tracking network consists of six transformer blocks. Within the tracking network's transformer blocks, we innovate by replacing the standard cross-attention layer with the referring cross-attention layer, as introduced in (Zhang et al., 2023a). Additionally, we conduct experiments with the temporal refiner (Zhang et al., 2023a) over 160k iterations, specifically analyzing sequences of 15 consecutive frames to enhance tracking accuracy.

For efficient training, we adopt a staged approach where the segmentation network is trained first, followed by the tracking network with all other parameters frozen, promoting stability and efficiency in learning, as suggested by previous studies (Zhang et al., 2023a; Li et al., 2023a). Optimization is carried out using the AdamW optimizer (Loshchilov & Hutter, 2017), with a starting learning rate of 1e-4 and a weight decay of 5e-2. The training process spans 40k iterations for the segmentation network and 160k iterations for the tracking network, with learning rate reductions scheduled at 28k and 112k iterations, respectively. During training, we sample three frames for the segmentation network and five frames for the tracking network from each of eight batched videos. These frames undergo resizing to ensure the shorter side is between 320 and 640 pixels, while the longer side does not exceed 768 pixels. The loss function weights are set to $\lambda_{cls} = 2.0$, $\lambda_{bce} = 5.0$, $\lambda_{dice} = 5.0$, $\lambda_{ctx} = 2.0$, and $\lambda_{pro} = 2.0$ to balance the contributions of each component during training. For inference, the shorter side of input frames is scaled down to 448 pixels to maintain a consistent aspect ratio across inputs. All experiments are conducted using 8 RTX2080Ti GPUs for the ResNet-50 backbone and 8 RTX3090 GPUs for the Swin-L and ViT-L backbones, ensuring adequate computational resources are available for the demands of each model configuration.

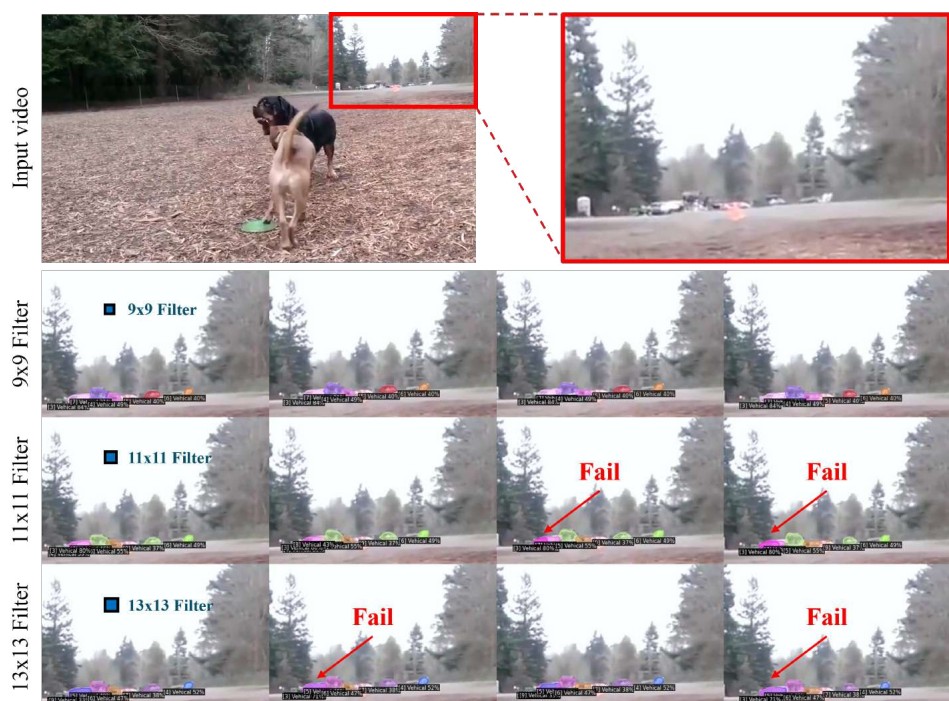

Figure 7: VIS results with various filter sizes.

## A.3 FURTHER STUDIES

**Analysis on object embeddings.** To demonstrate the effectiveness of our context-aware instance learning, we compare the distribution of object embeddings from three different models, as shown in Fig. 5. MinVIS does not engage in video learning, resulting in less effective distinction between objects. Compared to MinVIS, CTVIS shows a clearer object distinction by employing contrastive learning among object embeddings, but it still exhibits some overlaps in object clusters. In contrast, CAVIS forms much more distinct object clusters, highlighting the advantage of leveraging contextual information for object identification. This trends are reflected in the VIS results, as shown in Fig. 6.

**Comparison of PCC loss with VISAGE.** VISAGE employs contrastive loss on appearance features extracted from feature maps of the backbone encoder, which are also utilized for object matching during inference. This approach specifically aims to achieve more accurate object matching using appearance features. In contrast, our proposed PCC loss operates on feature maps extracted from the pixel decoder, targeting representation learning at the semantic level. To investigate whether VISAGE's appearance-level contrastive learning also contributes to representation learning, we conducted additional experiments. Using the same baseline architecture and a basic loss function, we tested PCC loss and VISAGE's appearance loss separately, as shown in Tab. 6. The results indicate that our method achieves a +1.7 AP gain over the baseline even without contrastive learning between object features. When combined with contrastive learning, it demonstrates further synergy, achieving 28.9 AP. In contrast, the appearance-level loss results in marginal performance improvements, with gains of only +0.5 AP and +0.1 AP in both cases. These results highlight that the proposed PCC loss facilitates the learning of object feature representations, distinguishing it from existing losses.

| Method | CL with $\hat{Q}$ | |
|---|---|---|
| | ✗ | ✓ |
| Baseline | 26.4 | 28.2 |
| with Appearance loss | 26.9 (+0.5) | 28.3 (+0.1) |
| with PCC loss | 28.1 (+1.7) | 28.9 (+0.7) |

Table 6: Appearance loss vs PCC loss.

**Effective filter size.** Videos often contain objects of varying sizes, and for smaller objects, using an excessively large context area can introduce noise, leading to inaccurate matching as shown in Fig. 7. To better understand this effect, we analyze the impact of different filter sizes to identify the optimal value. Our findings indicate that the overall trend remains consistent, regardless of variations in the number of frames used during training, as shown in Tab. 4-(b).

Table 7: Ablation studies on each component of CAVIS. (a-d) present the results from the segmentation network, while the others present those from the tracking network. "CL" denotes contrastive learning.

(a) Context-aware feature learning, PCC loss

| | CL with $\hat{Q}$ | $\mathcal{L}_{CTX}$ | $\mathcal{L}_{PCC}$ | AP |
|---|---|---|---|---|
| (i) | | | | 26.4 |
| (ii) | ✓ | | | 27.9 |
| (iii) | | ✓ | | 29.1 |
| (iv) | | | ✓ | 27.6 |
| (v) | ✓ | | ✓ | 28.3 |
| (vi) | | ✓ | ✓ | **29.5** |

(b) Context filter size

| Filter size | AP |
|---|---|
| 3 | 27.3 |
| 5 | 28.3 |
| 7 | 28.7 |
| 9 | **29.5** |
| 11 | 28.9 |

(c) Sampled frames

| # of frames | AP |
|---|---|
| 2 | 29.5 |
| 3 | **30.0** |
| 4 | 28.7 |

(d) Context filter type

| Metric | Context filter type | |
|---|---|---|
| | Average | Learnable |
| AP | **29.5** | 28.4 |

(e) Cross-Attention for $\mathcal{T}$

| Metric | Cross-Attention | |
|---|---|---|
| | $\hat{Q}$ | $Q$ |
| AP | 34.4 | **36.1** |

(f) Context alignment

| Metric | Context alignmnet | |
|---|---|---|
| | ✗ | ✓ |
| AP | 32.8 | **36.1** |

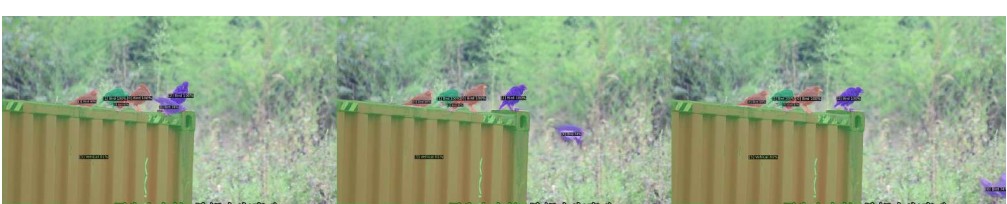

Figure 8: VIS results from our model on a video containing a fast-moving object.

**Robustness of our model.** Our method does not rely solely on context. By incorporating both context and instance features, our approach shows robustness even in scenes containing fast-moving objects where context changes rapidly, as shown in Fig. 8.

**Ablation study with minimal setups.** To simplify reproducibility, we additionally provide ablation studies on the OVIS dataset (Qi et al., 2022) with the ResNet-50 (He et al., 2016) backbone, detailed in Tab. 7. The results exhibit similar trends to those observed in Tab. 4, further validating the consistency of our findings. For these experiments, we train the segmentation network with 2 frames over 40k iterations, while the tracking network is trained with 5 frames over 40k iterations. Experiments (i-iii) show that implementing contrastive learning, whether with standard or context-aware instance features, leads to significant performance gains. Particularly, context-aware instance features result in a notable +2.7 AP improvement over the baseline, a considerable increase compared to the +1.3 AP improvement observed with standard instance features.

**Performance on long video.** We additionally report the perfor-
mance on the YouTube-VIS 2022 dataset, a well-known benchmark
featuring long video sequences. Its validation set includes 71 addi-
tional videos compared to the YouTube-VIS 2021 dataset, making
it particularly challenging due to the need for accurately tracking
dynamically appearing and disappearing objects over extended peri-
ods. We evaluate our model on these 71 long videos and compare it
against existing state-of-the-art models with a ResNet-50 backbone.
As shown in Tab. 8, our approach outperforms existing methods,
demonstrating that our context-aware modeling remains effective for robust object matching even in long-range video scenarios.

| Method | AP |
|---|---|
| MinVIS (Huang et al., 2022) | 23.3 |
| DVIS (Zhang et al., 2023a) | 31.6 |
| VITA (Heo et al., 2022) | 32.6 |
| DVIS++ (Zhang et al., 2023b) | 37.2 |
| GenVIS (Heo et al., 2023) | 37.5 |
| Ours | **38.6** |

Table 8: Comparison on YTVIS 2022 dataset.

**Additional qualitative results.** We provide additional qualitative results of CAVIS across various datasets, as depicted in Fig. 9-12. These results underscore the robust capability of CAVIS to track objects in diverse scenarios for both VIS and VPS tasks. Notably, CAVIS excels in environments featuring numerous similar objects, fast-moving objects, and significant occlusions, demonstrating its effectiveness across complex dynamic scenes.

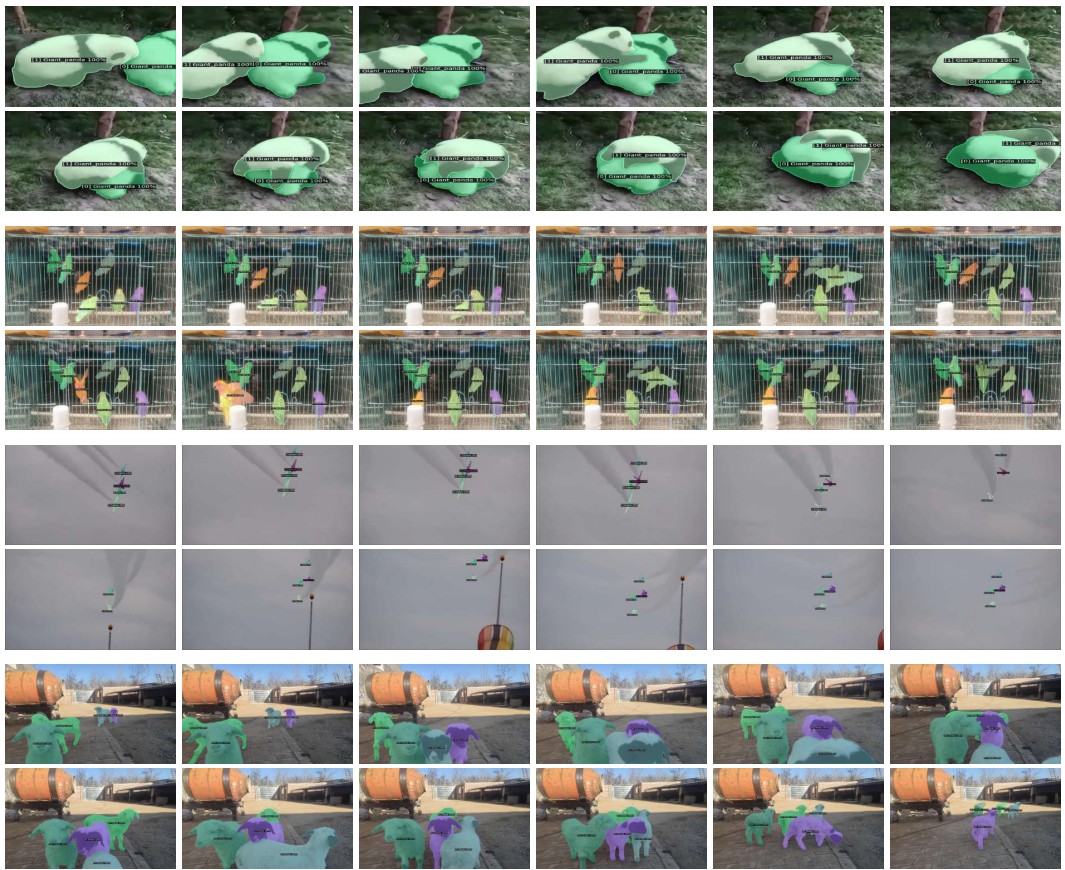

Figure 9: Additional qualitative results on OVIS dataset.

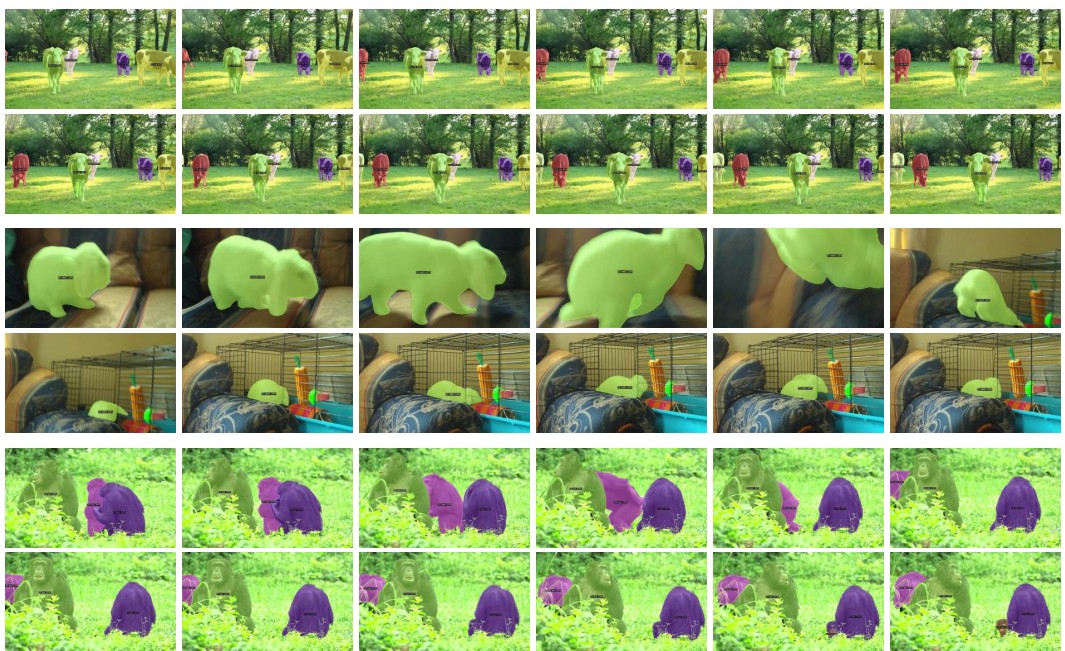

Figure 10: Additional qualitative results on Youtube-VIS 2019 dataset.

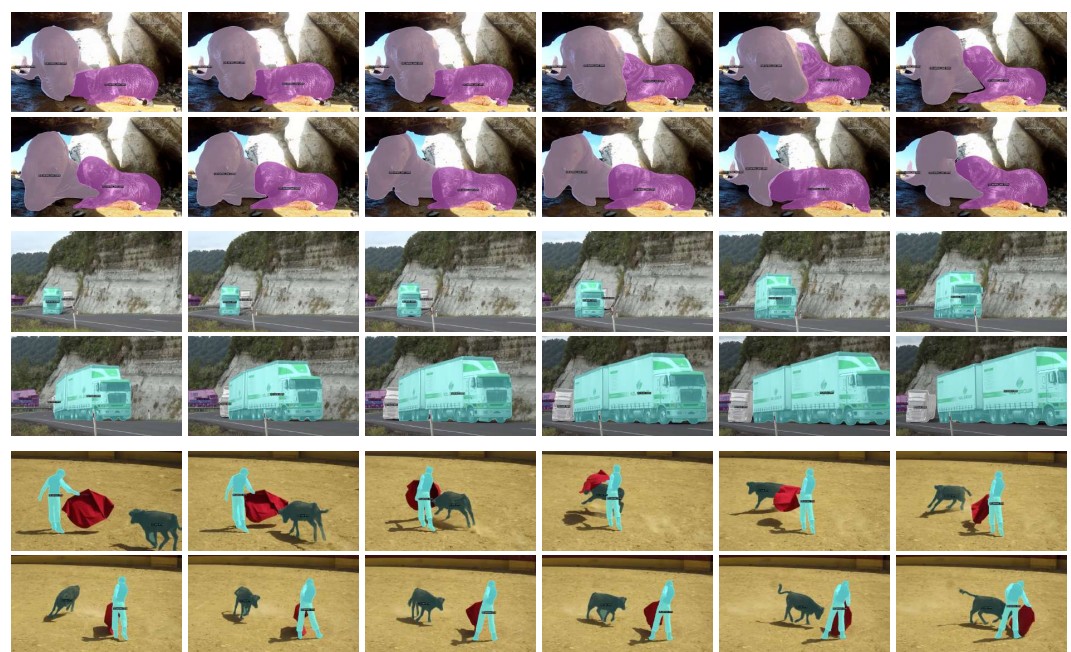

Figure 11: Additional qualitative results on Youtube-VIS 2021 dataset.

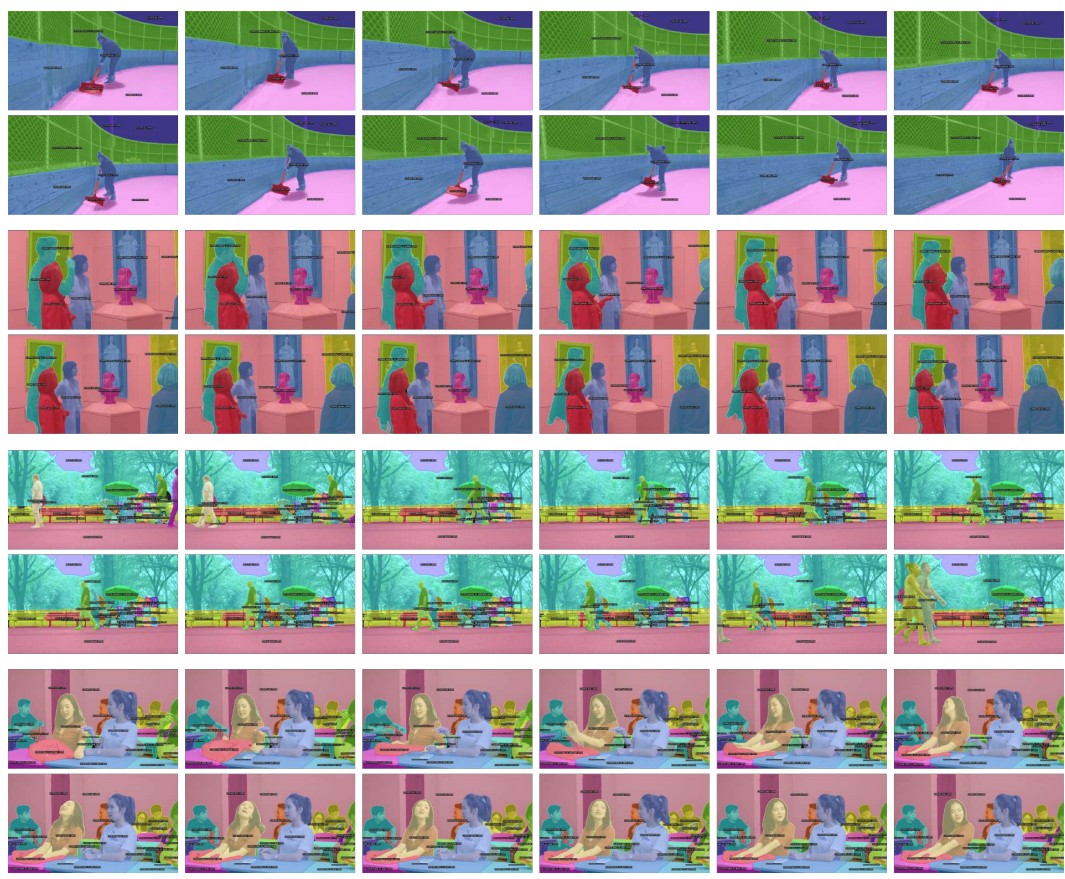

Figure 12: Additional qualitative results on VIPSeg dataset.

