# OpenReview forum: "Context-Aware Video Instance Segmentation"
_ICLR.cc/2025/Conference — Submitted to ICLR 2025_

### Official Review · Reviewer_Fndi · 2024-10-27

**Soundness:** 2
**Presentation:** 2
**Contribution:** 1
**Rating:** 5
**Confidence:** 5

**Summary:**

This paper addresses the video segmentation task and presents CAVIS, a method that enhances instance association by leveraging contextual information surrounding the instances. Specifically, the paper introduces the Context-Aware Instance Tracker (CAIT), which incorporates both surrounding features and instance features. The context feature encodes the “contextual information” related to each object. To further utilize this contextual information, the authors propose context-aware instance matching through a modified cross-attention module. Additionally, to improve the learning of instance matching, the paper introduces the Prototypical Cross-frame Contrastive (PCC) loss. With these components, the proposed method achieves state-of-the-art performance across all benchmarks.

**Strengths:**

This paper introduces a new perspective on the video segmentation task. The authors argue that better identification requires not only focusing on the object itself but also considering the contextual cues around it. This approach is well-motivated, and the experiments demonstrate the effectiveness of the proposed method.

**Weaknesses:**

**Major Weaknesses:**

**1. Insufficient Support for the Motivation**

The proposed components are insufficient to fully support the paper’s motivation. The surrounding feature is generated by extracting the edge area of the context feature, which aggregates nearby information. However, it still contains noisy data from small regions, making it unclear whether the surrounding feature provides meaningful “contextual information” that can effectively distinguish target instances. For example, in Fig. 1, the surrounding feature of the bicycle should ideally capture information about the person riding it, but the proposed approach seems to struggle with incorporating such essential contextual details. Moreover, the ablation study on context-aware feature learning is insufficient to support the motivation. Further analysis is required to demonstrate how the extracted surrounding feature can consistently capture relevant context and contribute to better instance distinction.

**2. Similarity to previous work**

The design of the PCC loss closely resembles the appearance-guided enhancement method introduced in VISAGE [1]. In VISAGE, the appearance query is generated using the same procedure described in Eq. 11. Moreover, the generated appearance query is employed for contrastive learning to ensure high appearance similarity, which aligns closely with the motivation behind PCC. Therefore, the proposed PCC method appears to be a re-implementation of VISAGE’s approach.


**Minor Weaknesses:**

**1. Missing Citations**

Some relevant works are omitted in the citations. As mentioned above, VISAGE [1] presents a similar learning objective and also incorporates additional information to enhance tracking quality by resolving ambiguities.

**2. Unconvincing Ablation Studies**

The ablation studies are not well-designed. Since the settings for the ablation experiments differ from those in the main experiments, it is difficult to assess the impact of each component on the final performance.

**3. Redundant Notations and Reduced Readability**

The paper contains many equations and redundant notations, which detract from its readability.


[1] Kim, Hanjung, et al. "VISAGE: Video Instance Segmentation with Appearance-Guided Enhancement." European Conference on Computer Vision. Springer, Cham, 2025.

**Questions:**

**Q1: What Information is Encoded in the Surrounding Feature?**

To address the weakness mentioned in Major.1, it is important to analyze what type of information the surrounding feature encodes. A useful experiment would be to plot the attention map of the surrounding feature to identify which areas are being attended to. This would help determine whether the surrounding feature captures meaningful contextual information.

**Q2: Do the Ablation Study Trends Hold Across Experiment Settings?**

If the settings of the ablation studies were aligned with those of the main experiments, would the overall performance trends remain consistent? This is crucial to confirm that the ablation studies reflect the real contribution of each component under consistent conditions.

**Q3: Why Does a Wider Context Filter Reduce Performance?**

The ablation studies show that increasing the context filter size degrades performance. Why does a wider filter (e.g., size 11) lead to worse results compared to a smaller one (e.g., size 9)? Plotting the difference in attention maps between filter sizes 9 and 11 could provide insights into how the filter size influences attention distribution and performance.

**Q4: Why Use the Instance Feature as the Value in Context-Aware Matching?**

In context-aware instance matching, the value is set to the instance feature rather than the context feature. Since the context feature already encompasses the instance feature, it seems more intuitive to use the context feature as the value. Has any analysis been conducted to explain this design choice? Exploring this further could clarify whether using the context feature would offer better performance.

---

> ### Author Response · Authors · 2024-11-20
> **Analysis on surrounding features (Response to Major Weaknesses 1, Questions 1.)**
>
> Our approach leverages surrounding context to improve object tracking, particularly in heavily occluded scenarios. For example, in [[this figure]](https://anonymous.4open.science/r/CAVIS_ICLR2025-73E7/rebuttal/more_predictions(5Yhw_Fndi).pdf), consider the "purple elephant'' in the leftmost part of the first frame.
> Pixels near its boundary overlap with the "light green elephant's rear.'' In the second frame, similar "light green elephant rear'' pixels remain nearby, enabling an easy match. In the third frame, pixels around the "purple elephant" are surrounded by pixels from the "light green elephant's rear and legs." This context persists across subsequent frames, enabling precise tracking using this information. In contrast, traditional methods like MinVIS and CTVIS, which rely solely on instance features, often fail under heavy occlusion. As shown in Table 1 of the main text, our method outperforms the conventional methods by a large margin, especially on the OVIS dataset.
> This difference is evident in the t-SNE visualization (see Fig. 4 in Appendix), where our context-aware modeling robustly distinguishes objects even in challenging occlusion scenarios.
>
> **Attention map.**
> We appreciate your suggestion to analyze the surrounding features through attention map visualization. However, these features are incorporated within the attention operations of CAIT, where an N$\times$N attention map is computed to represent object-matching relationships. Visualizing this map does not directly clarify the role of surrounding features. Instead, we have provided detailed analyses of model predictions and t-SNE visualizations (Fig 4, Appendix), which more effectively demonstrate the impact and benefits of our surrounding feature modeling.

---

> ### Author Response · Authors · 2024-11-20
> **Comparison to VISAGE (Response to Major Weaknesses 2 and Minor Weaknesses 1.)**
>
> VISAGE, proposed by Kim, Hanjung, et al., employs appearance features for object matching, specifically relying on low-level feature maps extracted from the backbone encoder. In contrast, our approach utilizes high-level feature maps derived from a pixel decoder, ensuring temporal consistency in the semantic features of the same object across a video. Moreover, these features are not utilized during inference in our method.
> Given the distinct motivations and differences in training and testing methodologies, our approach cannot be regarded as a simple re-implementation of VISAGE.
> Furthermore, our method demonstrates superior performance compared to VISAGE on the OVIS, YTVIS19, and YTVIS22 benchmarks as shown in Table 9.
> To provide additional context, we will include a comparison with VISAGE, along with the accompanying table, in the final version.
>
> **Table 9. VISAGE vs Ours**
> |        | OVIS     | YTVIS19   | YTVIS21   | YTVIS22   |
> |--------|:----------:|:-----------:|:-----------:|:-----------:|
> | VISAGE | 36.2     | 55.1      | **51.6**  | 37.5      |
> | Ours   | **37.6** | **55.7**  | 50.5      | **38.6**  |

---

> ### Author Response · Authors · 2024-11-20
> **Ablation study with the same setting of main experiments (Response to Minor Weaknesses 2 and Questions 2.)**
>
> When conducting the ablation study under the same setup as the main experiments, we observed similar trends. In the initial version of the paper, we presented results from a tracker model trained with fewer iterations (40k) to simplify reproducibility. To address potential confusion, we now report the ablation study results for the model trained with the full number of iterations (160k) in Table 10-12.
> In the final version, we will revise the structure to include the ablation study results from the main experimental setup in the main section, while relocating the minimal setup results to the appendix for reference.
>
>
> **Table 10. Detailed ablation study**
>
> |                          |        Segmenter         |          |           | Tracker |          |
> | :----------------------: | :----------------------: | :------: | :-------: | :-----: | :------: |
> | $\mathcal{L}_\text{CTX}$ | $\mathcal{L}_\text{PCC}$ |    AP    | $\hat{Q}$ |   $Q$   |    AP    |
> |                          |                          |   26.4   |     ✓     |         |   33.2   |
> |                          |            ✓             |   28.1   |     ✓     |         |   34.2   |
> |            ✓             |                          |   29.7   |     ✓     |         |   34.8   |
> |            ✓             |                          |   29.7   |           |    ✓    |   37.2   |
> |            ✓             |            ✓             | **30.0** |     ✓     |         |   35.3   |
> |            ✓             |            ✓             | **30.0** |           |    ✓    | **37.6** |
>
> **Table 11. Ablation study on filter type**
> | Metric | Average | Learnable |
> |--------|---------|-----------|
> | **AP** | **30.0** | 28.7      |
>
> **Table 12. Ablation study on context alignment**
> | Metric | ✗  | ✓  |
> |--------|---------------------------|-----------------------------|
> | **AP** | 33.2                      | **37.6**                   |

---

> ### Author Response · Authors · 2024-11-20
> **Clarification of notation and equations (Response to Minor Weaknesses 3.)**
>
> We have clarified the previously confusing notations in Table 13. This clarification will be reflected in the final version.
>
> **Table 13. Confusing notations**
> |   Symbol    | Description                             |   Symbol    | Description                              |
> | :---------: | --------------------------------------- | :---------: | ---------------------------------------- |
> |  $\hat{Q}$  | instance features                       |     $M$     | mask predictions                         |
> | $\tilde{Q}$ | instance surrounding features           | $\acute{M}$ | boundary scores processed from $M$       |
> |     $Q$     | context-aware instance features         |     $F$     | the last feature maps from pixel decoder |
> |   $Q^{*}$   | aligned context-aware instance features |  $\bar{F}$  | feature maps processed by average filter |

---

> ### Author Response · Authors · 2024-11-20
> **Trade-off between performance and filter size (Response to Questions 3.)**
>
> Videos often contain objects of varying sizes, and for smaller objects, using an excessively large context area can introduce noise, leading to inaccurate matching as shown in [[this figure]](https://anonymous.4open.science/r/CAVIS_ICLR2025-73E7/rebuttal/context_size(Fndi).pdf). To better understand this effect, we analyzed the impact of different filter sizes to identify the optimal value. Our findings indicate that the overall trend remains consistent, regardless of variations in the number of frames used during training or changes in the backbone architecture, as shown in Table 14-15.
>
>
> **Table 14. ablation study on filter size with R50 backbone**
> |                      |        |        |filter size|   |         |
> |------------------|:-------:|:-------:|:-------:|:-------:|:--------:|
> | # of frames   | **3** | **5** | **7** | **9** | **11** |
> | **2 Frames**     | 27.3  | 28.3  | 28.7  | 29.5  | 28.7   |
> | **3 Frames**     | 27.3  | 28.7  | 29.3  | **30.0** | 29.1   |
> | **4 Frames**     | 27.2  | 28.1  | 28.1  | 28.7  | 28.0   |
>
>
> **Table 15. ablation study on filter size with Swin-L backbone**
> |                      |        |        |filter size|   |         |
> |------------------|:-------:|:-------:|:-------:|:-------:|:--------:|
> | # of frames   | **3** | **5** | **7** | **9** | **11** |
> | **2 Frames**     | 39.4  | 39.8  | 40.7  | 40.8  | 40.1   |
> | **3 Frames**     | 39.7  | 40.2  | 40.8  | **41.2** | 40.9   |
> | **4 Frames**     | 39.4  | 40.1  | 39.8  | 39.8  | 40.0   |

---

> ### Author Response · Authors · 2024-11-20
> **Value in context-aware matching (Response to Questions 4.)**
>
> The concern that context-aware features, which include instance features, could be used as the value in context-aware matching is understandable. However, during segmenter training, the instance features specifically drive the segmentation prediction. Therefore, it is more effective to use instance features as the value for matching. Our experimental results also confirm that using instance features leads to better performance, as shown in Table 16. Thank you for highlighting this potential confusion. We will clarify this reasoning and present these findings in the final version.
>
> **Table 16. Ablation study on value feature**
> | value | $\hat{Q}$ | $Q$  |
> |-------|:-----------:|:------:|
> | AP    | 37.6      | 36.8 |

---

> > ### Comment · Reviewer_Fndi · 2024-11-21
> > **Unresolved Novelty Concern about PCC Loss**
> >
> > I have read the authors’ responses and the other reviewers’ reviews. However, I still have concerns about the PCC loss.
> >
> > ---
> >
> > In response to the concern raised by rJtA, the authors stated:
> > >However, as far as we know, no distinct approach applies contrastive loss at the pixel level across frames.
> >
> > This argument is incorrect, as VISAGE also applies contrastive loss at the pixel level (by extracting features corresponding to the mask) across frames.
> >
> > Moreover, apart from their use of these features during inference, PCC loss and VISAGE share the same motivation of **ensuring consistency in object-level features across frames**.  In VISAGE, the authors stated, _“we integrate a contrastive loss, which enhances the model’s ability to distinguish between instances across different frames”_, which is well-aligned with this motivation.
> >
> > Additionally, although benchmark performance cannot fully reflect their significance and the comparison may not be entirely fair, when comparing the impact of PCC loss in Table 10 with the performance of VISAGE (35.3 vs. 36.2), it is difficult to conclude that CAVIS demonstrates superior performance.
> >
> > ---
> > \* At ICLR, authors are allowed to revise their submitted paper during the discussion period, which helps reviewers identify the revised sections that address their concerns. However, in this case, the authors only promised revisions but did not incorporate them into their updated submission.

---

> ### Author Response · Authors · 2024-11-22
> **Response to unresolved concern and the revision**
>
> Thank you for pointing out the aspects we overlooked. We have uploaded a revised version of the paper, addressing your comments, including the response to your latest remarks on the comparison with VISAGE. Below is our detailed reply:
>
> **(1) Difference**
>
> VISAGE applies contrastive loss to low-level features extracted from the backbone and utilizes these features during inference. In contrast, our method applies contrastive loss to feature maps from the pixel decoder, **targeting semantic-level representation learning**. We conducted additional experiments with the same experimental setup of our ablation study (Table 4-(a) in our revised main text) to evaluate the effectiveness of VISAGE's loss in representation learning. As shown in Table 17., the appearance-level loss results in a minimal improvement of +0.5 AP over the baseline, whereas our method achieves a performance gain of +1.7 AP over the baseline. This demonstrates that our PCC loss is significantly more effective for representation learning. Therefore, VISAGE’s contrastive loss and our PCC loss differ fundamentally in both purpose and function. This distinction has been clarified in the Introduction section and the Appendix of our revised paper.
>
> **Table 17. Appearance loss vs PCC loss without CL**
> | Baseline | Appearance loss | PCC loss |
> |:----------:|:-----------------:|:----------:|
> | 26.4     | 26.9            | **28.1** |
>
> **(2) Performance**
>
> We apologize for any confusion caused by Table 10. The performance in **the 5th row (35.3)** corresponds to object matching using **instance features ($\hat{Q}$)**, while **the 6th row (37.6)** reflects matching with **context-aware instance features ($Q$)**. To avoid confusion, we have revised the main text to explicitly indicate whether context-aware matching is used (see Table 4-(a) in the main text). Additionally, **the 4th row of Table 10** shows that our method achieves **37.2 AP without relying on PCC loss**, which is 1.0 AP higher than VISAGE. This result highlights the superiority of our context-aware modeling over the VISAGE approach.

---

> > ### Comment · Reviewer_Fndi · 2024-11-23
> >
> > First, I would like to thank the authors for addressing my requests in their response and for providing the revised version. The revision clearly helps to identify the differences compared to the originally submitted version.
> >
> > In response to my concerns, the authors highlighted two aspects:
> > * Although VISAGE extracts low-level features from the backbone encoder and utilizes them during inference, CAVIS extracts semantic-level features from the pixel decoder and does not utilize them during inference.
> > * Compared to VISAGE’s Appearance loss, PCC loss performs well.
> >
> > However, these replies are insufficient to fully address the raised concerns:
> > 1. My concern was the authors’ incorrect response to reviewer rJtA. Since VISAGE also applies pixel-level contrastive loss, the authors’ response is incorrect, regardless of whether they were aware of this during the rebuttal process. However, this was not mentioned in the authors' responses.
> > 2. According to the authors, the main difference between VISAGE and CAVIS lies in the architectural design—specifically, the location of feature extraction (backbone vs. pixel decoder) and the type of representation encoded (low-level vs. semantic-level). If the authors wish to emphasize this difference in terms of the characteristics or motivation, further analysis is necessary. Table 17 seems to only present ablation studies on the architectural design, specifically regarding where the features are extracted. It does not appear to include supportive experiments highlighting the unique contribution of the PCC loss compared to VISAGE. Therefore, if the authors aim to claim a significant contribution for the PCC loss, additional analyses are essential.

---

> ### Author Response · Authors · 2024-11-24
> **Response to the comments on incorrent response and PCC loss**
>
> **1. Incorrect response**
>
> Thank you for pointing out our incorrect response to Reviewer rJtA's comments. Following your feedback, we have explicitly addressed the updates made in response to Reviewer rJtA's comments.
>
> **2. Distinctiveness of PCC loss**
>
>  To clearly demonstrate the distinctiveness of the PCC loss, we conducted additional experiments and revised Sections 4.2 of the main paper and A.3 of the appendix. Our response is as follows:
>
> In VIS methods, object feature representation learning is crucial because tracking relies on matching object features across frames. Existing approaches have primarily addressed this by employing contrastive learning between object features ($\hat{Q}$). **These object features** predict masks based on the similarity measure with **the feature map** extracted from the pixel decoder. In other words, object features and the pixel decoder's feature map are **semantically intertwined**. Consequently, ensuring pixel-level consistency across feature maps between frames can **further support the learning of robust object representations.**
>
> Experimental results in Table 19 demonstrate that while the appearance loss shows minimal improvement, the PCC loss exhibits strong synergy with contrastive learning. (Table 17 highlights the effectiveness of the PCC loss in representation learning even without contrastive learning, further validating its unique contribution to improving object feature representations.)
>
>
> **Table 19. Synergy with the standard contrastive learning**
> | Method                        | AP         |
> |-------------------------------|------------|
> | baseline + CL                | 28.2       |
> | baseline + CL + Appearance loss     | 28.3 (+0.1)|
> | baseline + CL + PCC loss     | **28.9 (+0.7)**|

---

> > ### Comment · Reviewer_Fndi · 2024-11-24
> >
> > After contrastive learning was first adopted in video tracking tasks for temporal consistency in QDTrack [1], many subsequent works have improved upon it in various ways:
> >
> > * IDOL [2] and Video K-Net [3] integrated contrastive learning into query-based methods. IDOL simplified sample selection by leveraging the Optimal Transport problem, while Video K-Net extended its frame-level baseline to video-level using contrastive learning.
> > * CTVIS [4] followed IDOL’s formulation but enhanced contrastive learning reliability by introducing a memory bank and adding noise to the memory bank updates for robust learning.
> > * Tube-Link [5] introduced temporal contrastive learning and, to align with its tube-level architecture, performed contrastive learning at the clip level instead of the frame level.
> > * VISAGE proposed an additional loss, expanding beyond the sole use of traditional embedding loss, by introducing Appearance Loss and combining it with embedding loss. Appearance Loss differs from conventional embedding loss by extracting features from feature maps using masks, rather than relying solely on queries.
> >
> > These previous works demonstrate distinctive perspectives and unique contributions compared to their predecessors. However, PCC Loss does not exhibit any fundamentally unique innovations.
> >
> > As I have consistently pointed out, it is challenging to interpret the independent contribution of PCC Loss. Even after reviewing the additional experiments provided by the authors, PCC Loss appears to be more of an architectural design choice for appearance loss rather than a novel conceptual advancement.
> >
> > The motivation for PCC Loss seems orthogonal to the primary motivation of CAVIS, which is **utilizing context-aware information for video instance segmentation**. Since PCC Loss is an off-the-shelf design for CAVIS, its distinct contribution needs to be clearly demonstrated. If PCC Loss shares the same motivation of leveraging context-aware information, the authors should provide evidence of its effectiveness in this context. For instance, they could evaluate how the choice of loss affects the model’s context-aware ability and include experiments beyond conventional benchmarks that may not fully reflect context-awareness.
> >
> > I fully agree that extracting features from the pixel decoder is an excellent fit for CAVIS and achieves superior performance compared to extracting features from the backbone encoder. However, the authors should moderate their claims regarding the contribution of PCC Loss.
> >
> > ---
> >
> > [1] Pang, Jiangmiao, et al. "Quasi-dense similarity learning for multiple object tracking." Proceedings of the IEEE/CVF conference on computer vision and pattern recognition. 2021.
> >
> > [2] Wu, Junfeng, et al. "In defense of online models for video instance segmentation." European Conference on Computer Vision. Cham: Springer Nature Switzerland, 2022.
> >
> > [3] Li, Xiangtai, et al. "Video k-net: A simple, strong, and unified baseline for video segmentation." Proceedings of the IEEE/CVF Conference on Computer Vision and Pattern Recognition. 2022.
> >
> > [4] Ying, Kaining, et al. "Ctvis: Consistent training for online video instance segmentation." Proceedings of the IEEE/CVF International Conference on Computer Vision. 2023.
> >
> > [5] Li, Xiangtai, et al. "Tube-Link: A flexible cross tube framework for universal video segmentation." Proceedings of the IEEE/CVF International Conference on Computer Vision. 2023.

---

> > > ### Author Response · Authors · 2024-11-24
> > >
> > > Thank you for your continuous and detailed feedback. Based on your suggestions, we have reviewed and reflected the following points in the revised version:
> > >
> > > (1) Our primary innovation lies in context-aware modeling, with PCC loss serving as a supportive component independent of this focus. It was not our intention to claim that PCC loss is as innovative as CAIT. To clarify this, we have explicitly addressed this point in the Introduction section.
> > >
> > > (2) While PCC loss shares similarities with VISAGE and may not be universally regarded as highly innovative, we believe that our motivation for this loss and its demonstrated effectiveness provide meaningful insights for future researchers. Therefore, we have included a detailed explanation in the appendix to share this perspective.
> > >
> > > We sincerely hope that our insights inspire future researchers to build upon this work and contribute to even better advancements. If there are still points that might be misunderstood or not adequately reflected in the paper, we kindly request your review once again.

---

> > > > ### Comment · Reviewer_Fndi · 2024-11-24
> > > >
> > > > After reviewing the authors’ responses, most of my concerns have been addressed. However, I have a few requests to further clarify the issues I raised:
> > > > * In the **Introduction section**, the contributions of this paper are numbered as three. However, contribution #2, regarding PCC Loss, is not substantial enough to stand as an independent contribution point, as previously discussed. It should either be merged with contribution #1 or removed entirely.
> > > > * The most relevant work, **VISAGE**, should be referenced in both the **Related Works** and **Method** sections. Since VISAGE also incorporates an additional cue (as mentioned in minor weakness #1), it should be addressed in the Related Works section. Additionally, in the Method section (4.2), VISAGE should be cited, as it employs a very similar approach to PCC Loss, differing only in the source feature map used.
> > > >
> > > > ---
> > > >
> > > > During the discussion period, most of my concerns were resolved, including the authors’ overclaim regarding PCC Loss. As a result of the discussion, it became clear that the contribution of PCC Loss is not comparable to CAIT, which weakens the paper. Although the proposed method demonstrates superior performance across all benchmarks, it still does not meet the standards of ICLR. Therefore, I have decided to increase my rating to borderline reject.

---

> > > > > ### Author Response · Authors · 2024-11-24
> > > > >
> > > > > We appreciate you sharing your perspective, but we hold a differing view on some of the comments. Recent top-tier conference papers, including VISAGE (ECCV24), primarily center around a main concept while optionally introducing supporting submodules, which may not always be as significant as the main contribution.
> > > > >
> > > > > Our study addresses occlusion, a critical issue in the VIS field, through context-aware modeling, achieving significant performance improvements. Notably, on the OVIS dataset, which contains a substantial amount of occlusion, we achieved state-of-the-art performance with a significant margin. Our comprehensive experiments and detailed analyses further underscore the strength of our contributions.
> > > > >
> > > > > Although we recognize that the contribution of PCC loss may not be as prominent, our ablation studies demonstrate its effectiveness as a supportive submodule. We believe that our findings are substantial enough for publication, offering contributions comparable to those of other top-tier conference papers.

---

### Official Review · Reviewer_5Yhw · 2024-10-29

**Soundness:** 2
**Presentation:** 3
**Contribution:** 2
**Rating:** 6
**Confidence:** 4

**Summary:**

This paper proposes a method called Context-Aware Video Instance Segmentation (CAVIS), which leverages contextual information around segmented instances to enhance instance differentiation across video frames. Specifically, CAVIS uses an average filter on feature maps and a Laplacian filter to extract surrounding features of each instance. By aggregating both instance features and surrounding context features during instance matching, the approach can aligns instances across frames more accurately. Additionally, a Prototypical Cross-frame Contrastive (PCC) loss is employed to further enforce inter-frame instance consistency

**Strengths:**

1. The paper is well-organized and easy to follow.

2. The method is simple yet effective. Introducing contextual information to improve instance segmentation is a clear and logical approach.

3. Comprehensive experiments demonstrate the effectiveness of the model, which achieves state-of-the-art performance across various metrics when tested with different backbones

**Weaknesses:**

1. Provide a clear notation table or explicitly define each variant of Q when introduced. For example, there are multiple definitions for "Q" which make the equations confusing. For Eq. (6), specify the dimension over which the average is calculated.
2. Clarify why surrounding pixels are chosen as context and how this compares to using full context information. Explain the specific benefits of using surrounding pixels for improving segmentation. Ablation study to show individual effects of CAIT and PCC components.
3. More visualization in the ablation study could strengthen the paper. In addition to t-SNE visualizations, examples of segmentation that showcase the impact of Instance Surrounding Features and Context-Aware Instance Features would help illustrate the added benefits. For example, visualizations highlighting whether in-context features enhance instance segmentation precision, particularly at object boundaries, could clarify the practical effects of the proposed method like having side-by-side comparisons of segmentation results with and without the context-aware features, focusing on challenging cases like object boundaries or occluded instances.
4. How does the addition of context features affect the computational efficiency of CAVIS? How does the performance of CAVIS change as the number of frames increases and the context potentially changes more significantly over time?

**Questions:**

Please see the `Weaknesses’ above.

---

> ### Author Response · Authors · 2024-11-20
> **Clarification of notation and equations (Response to Weaknesses 1.)**
>
> We have clarified the previously confusing notations in Table 4. and will explicitly denote that the average filter is applied to the height and width dimensions. This clarification will be reflected in the final version.
>
>
> **Table 4. Confusing notations**
> |   Symbol    | Description                             |   Symbol    | Description                              |
> | :---------: | --------------------------------------- | :---------: | ---------------------------------------- |
> |  $\hat{Q}$  | instance features                       |     $M$     | mask predictions                         |
> | $\tilde{Q}$ | instance surrounding features           | $\acute{M}$ | boundary scores processed from $M$       |
> |     $Q$     | context-aware instance features         |     $F$     | the last feature maps from pixel decoder |
> |   $Q^{*}$   | aligned context-aware instance features |  $\bar{F}$  | feature maps processed by average filter |

---

> ### Author Response · Authors · 2024-11-20
> **Benefits of using surrounding features (Response to Weaknesses 2.)**
>
> An example of a model that utilizes full context is CAROQ, proposed by Choudhuri et al. in CVPR 2023. CAROQ defines the context feature as a memory bank of multi-level image features extracted by a pixel decoder (see Sec. 3.4 in CAROQ). This method uses features from all frames, which increases computational complexity and can result in lower performance. Moreover, in practical applications, CAROQ may encounter memory limitations due to its reliance on full-context features.
>
> Interestingly, such a full-context-dependent approach performs similarly to or worse than simpler models like MinVIS, which rely solely on instance features as shown in Table 5. In contrast, our method integrates surrounding features directly into the instance features, enabling efficient and effective matching. As illustrated in Figure 4 of the appendix, this strategy significantly enhances the distinction between instances within the video.
>
> **Table 5. CAROQ (full-context) vs other models**
> | Method  | OVIS     | YTVIS19   | YTVIS21   |
> |---------|----------|-----------|-----------|
> | CAROQ   | 25.8     | 46.7      | 43.3      |
> | MinVIS  | 25.0     | 47.4      | 44.2      |
> | Ours    | **37.6** | **55.7**  | **50.5**  |

---

> ### Author Response · Authors · 2024-11-20
> **Detailed ablation study on PCC and CAIT (Response to Weaknesses 2.)**
>
> We provide a detailed performance analysis by reporting the AP in Table 6. for both the segmenter and tracker when each component is applied, focusing on the impact of the PCC loss and CAIT components. Following the suggestion of Reviewer Fndi, all experiments were conducted under the same setup as the main experiments: training the segmenter with 3 frames over 40k iterations and training the tracker with 5 frames over 160k iterations.
>
>
> **Table 6. Detailed ablation study**
>
> |                          |        Segmenter         |          |           | Tracker |          |
> | :----------------------: | :----------------------: | :------: | :-------: | :-----: | :------: |
> | $\mathcal{L}_\text{CTX}$ | $\mathcal{L}_\text{PCC}$ |    AP    | $\hat{Q}$ |   $Q$   |    AP    |
> |                          |                          |   26.4   |     ✓     |         |   33.2   |
> |                          |            ✓             |   28.1   |     ✓     |         |   34.2   |
> |            ✓             |                          |   29.7   |     ✓     |         |   34.8   |
> |            ✓             |                          |   29.7   |           |    ✓    |   37.2   |
> |            ✓             |            ✓             | **30.0** |     ✓     |         |   35.3   |
> |            ✓             |            ✓             | **30.0** |           |    ✓    | **37.6** |

---

> > ### Comment · Reviewer_5Yhw · 2024-11-25
> >
> > Thank you for addressing my concerns and now the ablation study includes a more detailed analysis of the impact of different components.
> > I have read the authors’ responses and the other reviewers’ reviews and the corresponding response. After looking at the results of  PCC loss raises concerns about the contribution and agree with other reviewer ( Reviewer Fndi). I agree that the contribution of PCC Loss is not comparable to CAIT, which weakens the paper.

---

> ### Author Response · Authors · 2024-11-20
> **Comparison of VIS results with/without the context-aware features. (Response to Weaknesses 3.)**
>
> Aligning with the recent trend in VIS methodologies prioritizing tracking accuracy, our approach also focuses on enhancing tracking performance over segmentation accuracy. To achieve better object matching, we introduce instance surrounding features, providing a more effective matching method.
> We provide an additional comparison of VIS results and highlight the moments in the video where severe occlusion happens. (See [[this figure]](https://anonymous.4open.science/r/CAVIS_ICLR2025-73E7/rebuttal/more_predictions(5Yhw_Fndi).pdf)) Our model accurately tracks objects during occlusion by utilizing the surrounding context. In contrast, instance-centric models struggle with severe occlusion, often failing to track objects accurately.

---

> ### Author Response · Authors · 2024-11-20
> **Computational efficiency (Response to Weaknesses 4.)**
>
> We compared the inference speed of our approach against recent state-of-the-art methods, GenVIS and DVIS, to evaluate the computational cost. The inference speeds were measured under identical conditions on a 2080ti GPU. As shown in Table 7., our method requires an additional time cost of 5.5ms and 6.7ms compared to GenVIS and DVIS, respectively. However, this cost is justified by the performance gains of +5.7AP and +4.5AP, demonstrating a reasonable trade-off between increased computation and improved accuracy.
>
> **Table 7. Trade-off between inference speed and performance**
> | Method | Time (ms) | YTVIS19 (AP) |
> | ------ | :-------: | :----------: |
> | GenVIS |   80.1    |     50.0     |
> | DVIS   |   78.9    |     51.2     |
> | Ours   |   85.6    |     55.7     |

---

> ### Author Response · Authors · 2024-11-20
> **Performance on long video (Response to Weaknesses 4.)**
>
> Our approach achieves superior performance compared to existing methods on YTVIS22, a well-known dataset featuring long video sequences as shown in Table 8. While long-term consistency was not the primary focus of our method, we will include these results as a reference in the final version.
>
> **Table 8. Comparison on YTVIS22 dataset with R50 backbone.**
> | Method  | AP         |
> |---------|------------|
> | MinVIS  | 23.3       |
> | DVIS    | 31.6       |
> | VITA    | 32.6       |
> | DVIS++  | 37.2       |
> | GenVIS  | 37.5       |
> | Ours    | **38.6**   |

---

> ### Author Response · Authors · 2024-11-20
> **Predictions on fast-moving objects (Response to Weaknesses 4.)**
>
> Please note that our method does not rely solely on context. By incorporating both context and instance features, our approach shows robustness even in scenes containing fast-moving objects where context changes rapidly. (See [[this figure]](https://anonymous.4open.science/r/CAVIS_ICLR2025-73E7/rebuttal/fast_moving(5Yhw).pdf))

---

> ### Author Response · Authors · 2024-11-25
>
> Dear Reviewer 5Yhw,
>
> Thank you for taking the time to review our work and for providing insightful comments and feedback. Your input has been instrumental in helping us refine our research.
>
> We have carefully addressed the concerns you raised and included updated results and detailed responses for each point. We hope our revisions clarify the key issues you highlighted, but we are happy to engage in further discussion to ensure all concerns are thoroughly addressed. Please feel free to share any additional questions or areas where further clarification may be needed. Your feedback remains invaluable in strengthening our work.
>
> Thank you again for your thoughtful review. We look forward to furthering this constructive discussion.

---

### Official Review · Reviewer_Ln3A · 2024-10-31

**Soundness:** 3
**Presentation:** 3
**Contribution:** 3
**Rating:** 6
**Confidence:** 4

**Summary:**

This paper proposes a context-aware model for video instance segmentation. Context-aware features are extracted and used for instance matching. Prototypical cross-frame contrastive loss is introduced to reduce computational cost and maintain pixel consistency. Experiments on multiple public benchmarks demonstrate the effectiveness of the proposed method.

**Strengths:**

1.The motivation of utilizing context information for VIS to improve training accuracy makes sense.

2.The implementation of context-aware feature extraction and instance matching is reasonable.

3.The prototypical cross-frame contrastive loss improves cross-frame pixel level consistency with affordable computational cost.

4.Experiments on multiple benchmarks demonstrate the effectiveness of the proposed method. It outperforms existing works across these benchmarks.

**Weaknesses:**

1.Table 3(d) shows that learnable filter performs worse than average filter. Although the authors explain that it's because the average filter is well-defined, but the learnable filter is not. The reviewer is still curious about the result, and more analysis should be provided. For example, will other filters, like maximum or median filter generate similar performance as the average filter?

2.The results on the more recent dataset YouTube-VIS 2022 is missing. It contains more videos than previous YouYube-VIS 2019 and 2021 benchmarks. It would be good if the experimental results on this dataset are provided.

**Questions:**

Please check the Weaknesses part.

**Details Of Ethics Concerns:**

N/A.

---

> ### Author Response · Authors · 2024-11-20
> **Other filter types (Response to Weaknesses 1.)**
>
> As suggested by Reviewer Fndi, we conducted additional experiments comparing the performance of max and median filters on the surrounding features. All experiments in Table 2. followed the same setup as the main experiments: training the segmenter with 3 frames over 40k iterations.
> The results show that the average filter consistently outperforms both max and median filters, further supporting its effectiveness in our approach.
>
>
> **Table 2. Further study on filter type.**
> | Filter type     | AP        |
> |-----------------|-----------|
> | Average (Ours)  | **30.0**  |
> | Max             | 29.4      |
> | Median          | 29.6      |

---

> ### Author Response · Authors · 2024-11-20
> **Performance on YTVIS22 (Response to Weaknesses 2.)**
>
> Since YTVIS22 differs from YTVIS21 only in its test set, the results can be easily verified using the provided pretrained model. As shown in Table 3., our approach also achieves superior performance compared to existing methods on YTVIS22. While long-term consistency was not the primary focus of our method, we will include these results as a reference in the appendix of the final version.
>
> **Table 3. Comparison on YTVIS22 dataset with R50 backbone.**
> | Method  | AP         |
> |---------|------------|
> | MinVIS  | 23.3       |
> | DVIS    | 31.6       |
> | VITA    | 32.6       |
> | DVIS++  | 37.2       |
> | GenVIS  | 37.5       |
> | Ours    | **38.6**   |

---

> ### Author Response · Authors · 2024-11-25
>
> Dear Reviewer Ln3A,
>
> Thank you for taking the time to review our work and for providing insightful comments and feedback. Your input has been instrumental in helping us refine our research.
>
> We have carefully addressed the concerns you raised and included updated results and detailed responses for each point. We hope our revisions clarify the key issues you highlighted, but we are happy to engage in further discussion to ensure all concerns are thoroughly addressed. Please feel free to share any additional questions or areas where further clarification may be needed. Your feedback remains invaluable in strengthening our work.
>
> Thank you again for your thoughtful review. We look forward to furthering this constructive discussion.

---

### Official Review · Reviewer_rJtA · 2024-11-08

**Soundness:** 2
**Presentation:** 2
**Contribution:** 2
**Rating:** 3
**Confidence:** 4

**Summary:**

The paper introduces a video instance segmentation framework focused on improving instance tracking in complex video scenarios by integrating contextual information for each instance. The authors claim two key contributions: (i) a Context-Aware Instance Tracker (CAIT) that explicitly incorporates contextual information for each instance, and (ii) a Prototypical Cross-frame Contrastive (PCC) loss to improve the consistency of enriched features across frames, further enhancing tracking robustness. Experiments are performed on the YouTube-VIS 2019, 2021, OVIS, and VIPSeg datasets.

**Strengths:**

(i) Comprehensive experiments across several video segmentation datasets.
(ii) The paper is easy to follow.
(iii) The authors have shared the code.

**Weaknesses:**

Weaknesses:


(i) The authors should more clearly explain the novel contributions of the proposed approach compared to closely related work, such as (Choudhuri et al., 2023). Currently, the authors mention that “Choudhuri et al., 2023 introduces a context-aware relative query which reflects global and local context by aggregating multi-level image features from consecutive frames” without explicitly explaining how the contextual information captured by the proposed method is superior. Additionally, the authors are expected to include a comparison with Choudhuri et al., 2023, and recent approaches in the state-of-the-art comparison table. Similarly, the novel contributions of PCC compared to closely related loss formulations also need to be clarified.
Reference: Choudhuri et al., 2023: Context-Aware Relative Object Queries to Unify Video Instance and Panoptic Segmentation, CVPR 2023.

(ii) It is unclear whether the performance gain is due to the additional number of duplicate queries or from capturing contextual information, as claimed. For example, some conventional object detection methods, such as GroupDETR (ICCV 2023) [https://arxiv.org/abs/2207.13085], demonstrate the benefits of using additional object queries.

(iii) The method seems somewhat ad-hoc to me . For instance, it involves obtaining instance-level masks for each frame using Mask2Former, then using edge-filtered instance masks to capture surrounding features of instances, which are subsequently leveraged to improve mask prediction and tracking. Would performance suffer if the initial masks obtained from Mask2Former were inaccurate? In cases of heavy occlusion, I assume that temporal information from previous frames would be beneficial for accurate mask prediction as well. However, since the initial masks are predicted independently by Mask2Former, could this affect overall performance?

(iv) It appears that the proposed method mainly targets improving tracking or association performance, rather than mask quality. If so, why isn’t the proposed approach evaluated on multi-object tracking datasets like MOTS2020 and KITTI-MOTS in addition to the VIS datasets, similar to Choudhuri et al.?

**Questions:**

See the weaknesses mentioned above.

---

> ### Author Response · Authors · 2024-11-20
> **Novelty (Response to Weaknesses 1.)**
>
> **1. Comparison to CAROQ.**
>
> CAROQ, proposed by Choudhuri *et al.*, defines the context feature as a memory bank of multi-level image features extracted by a pixel decoder (see Sec. 3.4 in CAROQ). This approach uses features from all frames indiscriminately, which increases complexity and may lead to lower performance. Moreover, it can cause memory issues in practical applications due to memory limitations. In contrast, our method adopts a memory-efficient approach by focusing on the surrounding features of each object during tracking, enabling effective object matching between frames with reduced memory overhead. While both methods utilize the term "context", our definition and application of context differ fundamentally. Thus, CAROQ’s methodology is only tangentially related to our work.
>
> To further clarify the distinction, the below table highlights the superior performance of our approach over CAROQ on multiple benchmarks, demonstrating the effectiveness of our method.
>
> **Table 1. CAROQ vs Ours.**
> | Method | OVIS     | YTVIS19  | YTVIS21  |
> | ------ | -------- | -------- | -------- |
> | CAROQ  | 25.8     | 46.7     | 43.3     |
> | Ours   | **37.6** | **55.7** | **50.5** |
>
> **2. PCC loss.**
>
> VIS methods, including IDOL, CTVIS, and DVIS++, typically apply contrastive loss between instance features to enhance temporal consistency. However, as far as we know, no distinct approach applies contrastive loss at the pixel level across frames. The ablation study comparing a traditional contrastive loss with our PCC loss is reported in Table 3-(a) of our main text. If you were referring to another closely related loss, it would be helpful if you could provide the relevant reference.
>
> **(Updated)**
> Following Reviewer Fndi's comment, we have added an explanation in the revised version to clarify the differences between the loss used in VISAGE (ECCV'24) and the PCC loss.

---

> ### Author Response · Authors · 2024-11-20
> **Number of queries (Response to Weaknesses 2.)**
>
> We apologize for any confusion regarding the concatenation operation in Eq. (7). To clarify, we concatenate the instance feature $\hat{Q}$ and surrounding features $\tilde{Q}$ along the channel dimension, resulting in an $N \times 2C$ representation. This is then processed by an MLP to map it back to $N \times C$. Importantly, the number of queries remains unchanged throughout this process. Any observed performance improvement is therefore not attributable to an increase in the number of queries but rather to the effective integration of instance and surrounding features. We will update the main text to ensure this distinction is clearly explained.

---

> ### Author Response · Authors · 2024-11-20
> **Inaccurate mask predictions (Response to Weaknesses 3.)**
>
> Our model is designed to improve tracking accuracy by achieving precise object matching across frames rather than focusing on segmentation performance. Consequently, if Mask2Former produces inaccurate segmentation results, performance may decrease. However, even in scenarios with imprecise mask predictions, our proposed context-aware modeling can robustly track objects, as demonstrated in [[this figure]](https://anonymous.4open.science/r/CAVIS_ICLR2025-73E7/rebuttal/inaccurate_mask(rJtA).pdf).

---

> > ### Comment · Reviewer_rJtA · 2024-12-03
> > **Thank you for the rebuttal**
> >
> > I would like to thank the authors for the rebuttal. I also thank Reviewer Fndi for raising concerns regarding the authors' rebuttal related to the PCC loss. The rebuttal and subsequent discussions have addressed most of my concerns. However, a major concern remains: why does a video instance segmentation method focus solely on the association aspect instead of jointly addressing the segmentation and association problem? Furthermore, since the proposed method specifically targets the association problem, I still believe that evaluating the approach on MOT benchmarks, similar to Choudhuri et al., could better demonstrate the merits of the proposed association strategy.

---

> ### Author Response · Authors · 2024-11-20
> **MOT (Response to Weaknesses 4.)**
>
> We adhere to the evaluation criteria and dataset settings used in the most recent VIS works, including GenVIS, DVIS, CTVIS, TCOVIS, and VISAGE. Consistent with these approaches, our work focuses on Video Instance Segmentation (VIS), specifically enhancing tracking performance by leveraging prediction masks. As a result, we have not conducted comparisons on MOT datasets, as they fall outside the primary scope of our work.

---

> ### Author Response · Authors · 2024-11-24
> **Correction of inaccurate response regarding the PCC loss.**
>
> Apologies for any potential confusion caused. We initially misunderstood and mistakenly responded under the assumption that there were no losses similar to the PCC loss. In the revised version, we have addressed the issue raised by Reviewer Fndi regarding the similarity between VISAGE and the PCC loss. The updated explanation is as follows:
>
> **VISAGE** employs contrastive loss on **appearance features extracted from feature maps of the backbone encoder**, which are also utilized for object matching during inference. This approach specifically aims to achieve more accurate object matching using appearance features. In contrast, our proposed **PCC loss** operates on **feature maps extracted from the pixel decoder**, targeting **representation learning at the semantic level**. To investigate whether VISAGE's appearance-level contrastive learning also contributes to representation learning, we conducted additional experiments. Using the same baseline architecture and a basic loss function, we tested PCC loss and VISAGE’s appearance loss separately, as shown in the Table 18. The results indicate that our method achieves a +1.7 AP gain over the baseline even without contrastive learning between object features. When combined with contrastive learning, it demonstrates further synergy, achieving 28.9 AP. In contrast, the appearance-level loss results in marginal performance improvements, with gains of only +0.5 AP and +0.1 AP in both cases. These results highlight that the proposed PCC loss facilitates the learning of object feature representations, distinguishing it from existing losses.
>
> **Table 18. Appearance loss (VISAGE) vs PCC loss**
>
> | Method                  | without CL | CL with $\hat{Q}$ |
> |-------------------------|-----------------------|-----------------------|
> | Baseline               | 26.4                 | 28.2                 |
> | with Appearance loss   | 26.9 *(+0.5)*        | 28.3 *(+0.1)*        |
> | with PCC loss          | 28.1 ***(+1.7)***    | 28.9 ***(+0.7)***    |

---

> ### Author Response · Authors · 2024-11-25
>
> Dear Reviewer rJtA,
>
> Thank you for taking the time to review our work and for providing insightful comments and feedback. Your input has been instrumental in helping us refine our research.
>
> We have carefully addressed the concerns you raised and included updated results and detailed responses for each point. We hope our revisions clarify the key issues you highlighted, but we are happy to engage in further discussion to ensure all concerns are thoroughly addressed. Please feel free to share any additional questions or areas where further clarification may be needed. Your feedback remains invaluable in strengthening our work.
>
> Thank you again for your thoughtful review. We look forward to furthering this constructive discussion.

---

> ### Author Response · Authors · 2024-12-04
> **Final response**
>
> **(1) Main Focus of VIS**
>
> Recent advancements in segmentation architectures have significantly improved the quality of segmentation masks. However, tracking objects in dynamic videos remains a challenging task. As a result, the VIS field has shifted its focus from refining segmentation quality to enhancing tracking accuracy. To address this, we developed a framework specifically designed for videos with frequent occlusions, aiming to achieve more reliable tracking.
>
> **(2) Evaluation on MOT**
>
> Our research builds on the latest VIS methods, particularly those employing Mask2Former, as shown in the code we submitted. Given the segmentation-oriented nature of this baseline, conducting experiments on MOT benchmarks within the limited discussion period proved challenging. However, based on your suggestion, we tested the effectiveness of our context-aware association strategy by incorporating it into MOTRv2 (CVPR’23), a state-of-the-art MOT model.
> We conducted experiments on the DanceTrack benchmark. Since this dataset does not provide segmentation masks, we generated edge maps for the entire image using Sobel filters (see [[this figure]](https://anonymous.4open.science/r/CAVIS_ICLR2025-73E7/rebuttal/mot(rJtA).pdf)) and derived boundary maps for individual objects from the predicted bounding boxes. As shown in Table 19, our method outperforms the baseline on the benchmark. These results highlight the versatility of our approach, demonstrating its potential to be applied beyond video segmentation tasks like VIS and VPS, extending its effectiveness to MOT as well.
>
> **Table 19. Comparison on DanceTrack val**
> | Method       | HOTA        |
> |--------------|-------------|
> | MOTRv2       | 64.5        |
> | + context     | **65.1 (+0.6)** |

---

### Author Response · Authors · 2024-11-20

We thank all reviewers for their constructive comments. We appreciate the positive feedback on the effectiveness of the proposed method (Ln3A, 5Yhw), strong motivation behind the approach (Fndi), comprehensive experiments (rJtA, Ln3A, 5Yhw), and good readability (rJtA, 5Yhw). To effectively address the reviewers' questions, we have uploaded additional visualization materials to the [[annonymous url]](https://anonymous.4open.science/r/CAVIS_ICLR2025-73E7/rebuttal/).

---

> ### Author Response · Authors · 2024-11-22
>
> We have uploaded the revised paper based on the feedback from the discussion. We are truly grateful for the valuable comments provided by all the reviewers, which have significantly improved the quality of our paper. If there are any further suggestions or feedback during the remaining discussion, we will carefully consider them and make any necessary updates. We sincerely appreciate your efforts.
>
> The revised sections are highlighted in **Plum** color and are as follows:
>
> 1. **Comparison with Previous Works**
>    1-(1) We added an explanation of the differences between PCC loss and VISAGE in the introduction section and appendix. (rJtA, Fndi)
>    1-(2) We included the differences with CAROQ in the related works section. (rJtA)
>
> 2. **Method Section Organization**
>    2-(1) We added a table for the confusing notations. (5Yhw, Fndi)
>    2-(2) We provided an explanation of the dimensions of the operations in Eq. (6) and (7). (rJtA, 5Yhw)
>
> 3. **Updated Experimental Results and Analyses**
>    3-(1) We updated the performance of CAROQ and VISAGE in Table 2. (rJtA, Fndi)
>    3-(2) We updated the ablation study on other filter types (Ln3A)
>    3-(3) We updated the ablation study with the same experimental setup as the main table in Table 4, and moved the previous ablation study table to the appendix. (Fndi)
>    3-(4) We added an ablation study for the values used in context-aware matching. (Fndi)
>    3-(5) We added experiments on computational cost. (5Yhw)
>    3-(6) We added the following analyses in the appendix:
>    - Potential issues arising from inaccurate masks. (rJtA)
>    - VIS results of the t-SNE figure on the video. (5Yhw)
>    - Performance report on YouTube-VIS 2022. (Ln3A, 5Yhw)
>    - VIS results on videos with fast-moving objects. (5Yhw)
>    - Analysis of filter size. (Fndi)

---

### Meta-Review · Area_Chair_a8Zd · 2024-12-19

**Metareview:**

This paper proposes a video object segmentation framework by leveraging contextual information for each instance. Context-aware features are extracted and used for matching. Prototypical cross-frame contrastive loss is introduced to reduce computational cost and maintain pixel consistency. The experimental results on YTVIS and OVIS datasets are given.

The main strengths are 1) good performance and 2) well presentation.

The main weaknesses are 1) lack of novelty (the proposed method is similar to previous methods, like VISAGE), and 2) the contribution of PCC loss is not substantial.

After the author-reviewer discussion, the main issue of lack of novelty still remains ( recognized by Reviewers Fndi and rJtA). As this issue significantly degrades the contribution of this paper, the AC does not recommend it be accepted at this conference.
The authors are encouraged to consider the comments for their future submission.

**Additional Comments On Reviewer Discussion:**

In the initial comments, the concerns raised by the reviewers are lack of novelty (the proposed method is similar to previous works like VISAGE), 2) the contribution of PCC loss is not substantial, and 3) missing more detailed ablation studies and citations.

After the discussion, the authors provided more experimental results and clarifications on the contribution of this paper.
However, Reviewers Fndi and rJtA still have concerns about the limited contribution of this paper.
The final scores are 6, 6, 5, and 3.

---

### Decision · Program_Chairs · 2025-01-22

Reject